# Lung NR3C1$^+$ and CXCR6$^{high}$ T cells distinguish immunopathogenesis of human emphysema
Yun Zhang [1,2], Maor Sauler [3], David B. Corry [1,2,4,5], Scott A. Ochsner [6], Sarah Perusich[5], Li-Zhen Song[1], Joshua Malo[7], Raul San Jose Estepar [8], Francesca Polverino [1,2,9] & Farrah Kheradmand [1,2,4,5]

There is a significant knowledge gap in how T cells promote emphysema in smokers with chronic obstructive pulmonary disease (COPD). Single-cell RNA sequencing (scRNA seq) analysis of human samples and relevant clinical data can provide new mechanistic insights into disease pathogenesis. We generated a human lung scRNA seq dataset with extensive disease characteristic annotation and analyzed a second independent scRNA seq dataset to examine the pathophysiological role of T cells in emphysema. Comparisons of pulmonary immune landscapes in emphysematous (E)-COPD, non-emphysematous (NE)-COPD, and control showed positive enrichment of T cells in E-COPD. Pathway analyses identified upregulated inflammatory states in CD4 T cells as a distinguishing feature of E-COPD. Compared to controls, glucocorticoid receptor NR3C1 CD4 T cells were enriched in NE-COPD but were reduced in E-COPD. Interactions between macrophages and NR3C1$^+$ CD4 T cell subsets via CXCL signaling were strongly predicted in E-COPD but were absent in NE-COPD and control. The relative abundance of CD4 CXCR6$^{high}$ effector memory T cells positively correlated with preserved lung function in E-COPD but not in NE-COPD. These findings suggest that NR3C1$^+$ and CXCR6$^{high}$ effector memory subsets of CD4 T cells distinguish the immune-pathophysiological features of emphysema in human lungs. Targeting relevant T cell subsets in emphysema might provide new therapeutic opportunities.

Long-term cigarette smoking profoundly changes the transcriptomic landscapes in the lungs[1–3] and is directly linked to the development of chronic obstructive lung diseases (COPD)[4,5]. Studies in human smokers and experimental animal models of smoke-induced emphysema suggest that both the adaptive[6–10] and innate[11,12] immune systems contribute to the tissue destruction and severity of lung disease in COPD. Many human studies, however, classify COPD based on the degree of airway flow limitation, but the extent of lung tissue destruction, emphysema, which is the most clinically consequential outcome in smokers[13–17], is not reported. Under-reporting and excluding distinct subphenotypes have culminated in a poor

understanding of the immune-mediated pathophysiological changes in human emphysema.

Genome-wide association studies (GWAS)[18–20], proteomic[21,22], and transcriptomic[23,24] studies recognize emphysema as an independent disease phenotype in COPD. Notably, radiographic detection of emphysema can be discordant with the physiological manifestation of airflow obstruction[25–27]. These observations call for a better characterization of emphysema and COPD endotypes in preclinical and clinical studies.

The molecular mechanisms underlying the diverse clinical presentations of COPD remain less clear. Inflammation is strongly associated with

[1]Department of Medicine, Division of Pulmonary, Critical Care Medicine, and Sleep, Baylor College of Medicine, Houston, TX, USA. [2]Department of Pathology and Immunology, Baylor College of Medicine, Houston, TX, USA. [3]Department of Medicine, Yale University, New Haven, CT, USA. [4]Biology of Inflammation Center, Baylor College of Medicine, Houston, TX, USA. [5]Center for Translational Research on Inflammatory Diseases (CTRID), Michael E. DeBakey Department of Veterans Affairs, Houston, TX, USA. [6]Department of Molecular and Cellular Biology, Baylor College of Medicine, Houston, TX, USA. [7]Department of Medicine, Division of Pulmonary, Critical Care Medicine, and Sleep, University of Arizona, Tucson, AZ, USA. [8]Division of Radiology, Brigham and Women's Hospital, Harvard Medical School, Boston, MA, USA. [9]Department of Medicine, Asthma and Airway Disease Research Center, University of Arizona, Tucson, AZ, USA. ✉e-mail: francesca.polverino@bcm.edu; farrahk@bcm.edu

the development of COPD among tobacco smokers, and the emphysema variant of COPD is linked to the activation of the adaptive immune system[28]. Specifically, identification of oligoclonal autoreactive T cells in humans and experimental emphysema models[7,8,29] suggests a pathogenic role for antigens that can activate and expand immune cells in the lungs. Although the contribution of adaptive immunity in the pathophysiology of emphysema is widely acknowledged, how different types and states of T cells and their interactions with innate immune cells in the lungs emerge and promote disease development in emphysema remain unclear.

Despite significant efforts in identifying biomarkers that may predict disease outcomes in tobacco smokers, immune profiles in the peripheral blood have not always mirrored tissue immunity in the lungs[18,30]. This knowledge gap suggests the need to examine lung tissue cellular profiles to decipher how immune cells promote lung tissue destruction. Single-cell (sc)RNA sequencing has been revolutionary in characterizing the cellular landscape at a high-throughput scale in single-cell resolution[31], but large-scale studies that include exhaustively annotated clinical information are rare and have not focused on the role of T cells in human emphysema.

In this study, we analyzed scRNA sequencing data using a total of 108 human lung tissues. We used sixty-two human lung samples from well-characterized groups of smokers and non-smokers with extensive disease characteristic annotation and radiographic quantification of emphysema using percent low attenuation area to separate cases into three strata: emphysema predominant (E-COPD), non-emphysematous COPD (NE-COPD), and controls. We took advantage of the transcriptomic information of different immune cell types and cell states in the lungs to investigate the factors that may precipitate the emergence of pathogenic cell types across different COPD endotypes. A published scRNA sequencing dataset from 46 samples was used to validate the main results.

## Results

### Distinct human lung immune cell landscape in emphysema

We acquired human lung tissue samples from 62 individuals and performed scRNA sequencing (Fig. 1a). Patients were first stratified to COPD ($n = 37$) and Control ($n = 25$) based on airflow obstruction ($FEV_1/FVC$). COPD patients were further categorized based on computed tomography (CT)-based measurement of low attenuation area percentages (LAA%) below 950

Hounsfield units for each subject (Supplementary Fig. 1a–g, Table 1). We used the 5% LAA cutoff as the accepted threshold for the presence of emphysema[32]. Percent LAA separated the COPD cohort into non-emphysematous COPD (NE-COPD) ($n = 21$) and predominant emphysema (E-COPD) phenotypes ($n = 16$).

### Table 1 | Patient characteristics, in-house dataset one

| Variable | Control $N = 25$ | No Emphysema COPD $N = 21$ | Emphysema COPD $N = 16$ | p-value[a] |
|---|---|---|---|---|
| Age, Median (IQR) | 62 (52–72) | 72 (67–75) | 68 (66–68) | 0.034 |
| Sex, n (%) | | | | 0.062 |
| F | 14 (56) | 9 (43) | 3 (19) | |
| M | 11 (44) | 12 (57) | 13 (81) | |
| Smoke, n (%) | | | | <0.001 |
| Current | 5 (20) | 8 (38) | 0 (0) | |
| Former | 9 (36) | 12 (57) | 16 (100) | |
| Never | 11 (44) | 1 (4.8) | 0 (0) | |
| PackYear, Median (IQR) | 15 (0–25) | 28 (16–54) | 39 (20–52) | 0.001 |
| GOLD, n (%) | | | | <0.001 |
| Control | 25 (100) | 0 (0) | 0 (0) | |
| GOLD 0 | 0 (0) | 7 (33) | 0 (0) | |
| GOLD 1 | 0 (0) | 6 (29) | 2 (13) | |
| GOLD 2 | 0 (0) | 8 (38) | 2 (13) | |
| GOLD 4 | 0 (0) | 0 (0) | 12 (75) | |
| LAA %, Median (IQR) | 0 (0–1) | 1 (0–2) | 19 (16–30) | <0.001 |
| FEV1% predicted, Median (IQR) | 100 (94–100) | 77 (65–80) | 23 (19–34) | <0.001 |
| FEV1/FVC, Median (IQR) | 80 (74–80) | 67 (63–73) | 27 (22–58) | <0.001 |
| Steroids, n (%) | 1 (4.0) | 4 (19) | 10 (63) | <0.001 |

IQR interquartile range.
[a]Kruskal–Wallis rank sum test; Pearson's Chi-squared test; Fisher's exact test.

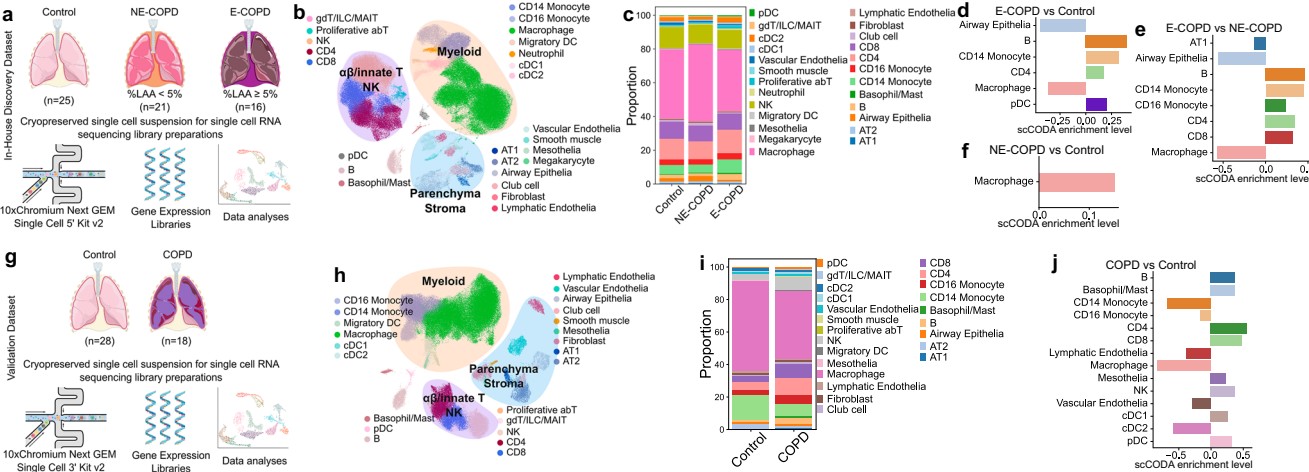

**Fig. 1 | Positive enrichment of T cells and negative enrichment of macrophages in Emphysematous COPD (E-COPD). a** Study design for in-house discovery dataset (**b–f**), and (**g**) for validation dataset (**h–j**). UMAP embedding of the global cellular landscape for the in-house discovery dataset (**b**) and the published validation dataset (**h**). Collective cellular proportions across disease groups in the in-house discovery dataset (**c**) and published validation dataset (**i**). Pairwise comparison of cell type differential abundance using scCODA in in-house discovery dataset between E-COPD vs. Control (**d**), E-COPD vs. NE-COPD (**e**), and NE-COPD vs Control (**f**).

Pairwise comparison of cell type differential abundance using scCODA in the published validation dataset between COPD and Control (**j**). The final parameter shown on the horizontal axis indicates degrees of enrichment. Positive values suggest positive enrichments, and negative values suggest negative enrichments. The false discovery rate (FDR) for scCODA differential abundance analyses was set at a threshold of 0.25. Artwork was generated from Bioicons (https://bioicons.com/) and NIH NIAID BioArt Source (https://bioart.niaid.nih.gov/). Modifications of the artwork were performed in Inkscape (https://inkscape.org/).

**Table 2 | Patient characteristics, validation dataset two**

| Variable | Control N = 28 | COPD N = 18 | p-value[a] |
|---|---|---|---|
| Gender, n (%) | | | 0.64 |
| Female | 12 (43) | 9 (50) | |
| Male | 16 (57) | 9 (50) | |
| Age, Median (IQR) | 47 (31–63) | 62 (58–66) | 0.002 |
| Race, n (%) | | | >0.99 |
| Asian | 1 (3.6) | 0 (0) | |
| Black | 1 (3.6) | 0 (0) | |
| Latino | 1 (3.6) | 0 (0) | |
| White | 25 (89) | 18 (100) | |
| Smoking, n (%) | 6 (21) | 17 (94) | <0.001 |

IQR interquartile range.
[b]Pearson's Chi-squared test; Wilcoxon rank sum test; Fisher's exact test.

A total of 48,569 cells from the control, 45,692 cells from NE-COPD, and 34,172 cells from the E-COPD group passed quality control. Twenty-four cell types were identified (Fig. 1b and Supplementary Fig. 1h). Immune cells represented over 90% of recovered cells, with macrophages (~48%) and αβT lymphocytes (~33%) together comprising over 80% of total cells (Fig. 1c). Other top immune populations include monocytes (~8%), NK (~10%), and cDCs (~4%). Top recovered non-immune components include alveolar epithelial cells (~2%) and vascular and lymphatic endothelial cells (~2%).

Because of the disparity of total cell counts and sample numbers in both COPD and control groups, we opted for the proportion-based Bayesian modeling approach, scCODA, for compositional analyses between different groups[33]. We compared cell-type enrichment distribution in E-COPD vs Control (Fig. 1d) and E-COPD vs NE-COPD (Fig. 1e). CD14 monocytes, B cells, and αβ CD4 T cells were positively enriched, whereas macrophages and epithelial cells showed negative enrichments in E-COPD compared to controls. In addition to the same cell populations, CD8 T cells were also enriched in E-COPD compared to NE-COPD, but this enrichment was not detected when compared to controls. In NE-COPD, macrophages showed positive enrichment when compared to controls (Fig. 1f). These findings indicate that the E-COPD phenotype has a distinctive immune profile from NE-COPD, suggesting that while adaptive immunity is highly associated with E-COPD, positive enrichment of macrophages is a characteristic feature in NE-COPD.

Because active smoking can strongly affect systemic and lung tissue immunity, we next examined its effect on immune cells. In the Control group, 20% of cases were current, 36% former, while 44% were never smokers (Table 1). In the NE-COPD group, 38% of cases were current, 57% former, while 5% were never smokers (Table 1). Because all patients in E-COPD were former smokers, we assessed the effect of smoking in the Control and NE-COPD groups. In the NE-COPD group, when comparing former smokers with never-smokers, macrophages were positively enriched, whereas CD8 and NK cells were negatively enriched (Supplementary Fig. 2a–c). In the same group, current smokers exhibited positive enrichment of pulmonary macrophages, but CD4 and CD8 T cells were negatively enriched (Supplementary Fig. 2). Consistently, comparing current smokers with former smokers, CD4 T cells were negatively enriched (Supplementary Fig. 2e). In the Control group, when comparing former smokers with never smokers, CD4 T cells were positively enriched, but macrophages were negatively enriched (Supplementary Fig. 2f–h). Current smokers, compared to former smokers, showed positive enrichment of CD8 T cells and macrophages but negative enrichment of CD16 monocytes (Supplementary Fig. 2i). Comparison between current and never smokers in the Control group did not yield any statistically significant results that passed the false discovery rate threshold.

We next used an independent and publicly available scRNA-seq dataset[34,35] that included end-stage COPD lung explant and rejected donor lung samples as controls to validate our findings (Fig. 1g,

Supplementary 3a–c, Table 2). Because information regarding %LAA was not available for the second COPD cohort, we could not stratify radiographically confirmed emphysema and were only able to compare end-stage COPD to controls. We found similar cellular enrichment patterns in the validation dataset with positive enrichments of the αβ CD4 and CD8 T lymphocytes, B cells, and negative enrichment of macrophages in end-stage COPD when compared to controls (Fig. 1h–j; Supplementary Fig. 3d).

Together, these two independent datasets provided evidence that the emphysema variant of COPD is characterized by the positive enrichment of αβ CD4 T cells and negative enrichment of pulmonary macrophages.

## Unique transcriptomic signatures of αβ CD4 T cells in emphysema

Because CD4 T cells were consistently positively enriched in E-COPD and end-stage COPD, we then set out to assess functional alternations in CD4 T cells using transcription factor activity analyses (Fig. 2a), gene set over representation analyses (ORA) (Fig. 2b), and gene set variation analysis (GSVA) (Fig. 2c–f). The transcription factor activity inference data showed increased STAT3, JUN, NFKB, and RELA activities in E-COPD compared to the NE-COPD and controls (Fig. 2a). STAT3 is a critical transcription factor downstream of IL6 signaling and can induce its expression[36–38]. Consistent with these findings, we found CD4 T cells in E-COPD exhibit upregulated IL2-STAT5 and IL6-JAK-STAT3 signaling by both ORA and GSVA analyses and E-COPD showed increased IL-17 production pathway in GSVA analysis compared to controls (Fig. 2b, e–g). Responses to chemokines and cytokines were also upregulated at an individual patient level in E-COPD compared to control or NE-COPD cohorts (Fig. 2c, d). IL6 induces expression of IL21 and IL23R in CD4 T cells, which are upstream of transcription factor, RORC, and IL17 expression[39]. STAT3 is also indispensable in the development of Th17 cells[40,41]. Notably, IL6-STAT3 pathways and IL-17 upregulation in CD4 in E-COPD further corroborate the pathogenic role of IL17/Th17 in human emphysema[42,43]. Furthermore, CD4 T cells in E-COPD also showed upregulation of inflammatory pathways, including interferon-gamma (IFN-γ), TNF, and IFN-α responses (Fig. 2b).

Together, these results suggest that augmented inflammatory pathways are a unique feature of CD4 T cells in the lungs of E-COPD and that the positive enrichments of CD4 T cells might be a result of increased responsiveness to chemokine and cytokine-mediated recruitment signals.

## Identification of distinct CD4 T cell subsets in human lungs

We next examined the heterogeneities of CD4 T cell subsets in the lung. Broadly, we identified 5 distinct subsets of CD4 T cells (Fig. 3a–d). Effector memory CD4 T cells were identified by the expression of S100A4 and IL32[44,45]. Naïve and central memory T cells were identified by the expression of CCR7, TCF7, CD62L (SELL), and LEF1, while naïve T cells were further distinguished by the absence of memory markers S100A4 and IL32[45]. Regulatory T (Treg) cells were identified by the concurrent expression of FOXP3, IL2RA, TIGIT, CTLA4, and IKZF2[46–49]. A subset of CD4 T cells expressing glucocorticoid receptor NR3C1 showed distinct gene expression profiles (Fig. 3b, c) and transcription factor activity (Fig. 3d), which we termed the NR3C1+ subset. To confirm the CD4 T cell subsets, we performed pseudotemporal analysis of the different populations using the Palantir modeling method[50]. As expected, naïve and central memory CD4 T cells exhibited the lowest levels of differentiation, whereas Tregs showed the highest level of differentiation (Fig. 3e–g). Notably, effector memory CD4 T cells exhibited an intermediate level of differentiation, whereas NR3C1-expressing CD4 T cells showed the second-highest levels of differentiation (Fig. 3e–g). We next confirmed the presence of distinct CD4 T cell subsets using an independent scRNA seq dataset (Supplementary Fig. 4a–c).

## Reduced relative abundance of NR3C1+ CD4 T subset in E-COPD

We next examined the relative abundance of glucocorticoid-receptor NR3C1-expressing T cells in the study cohorts. We found that the relative abundance of the NR3C1+ subset of CD4 was significantly reduced in

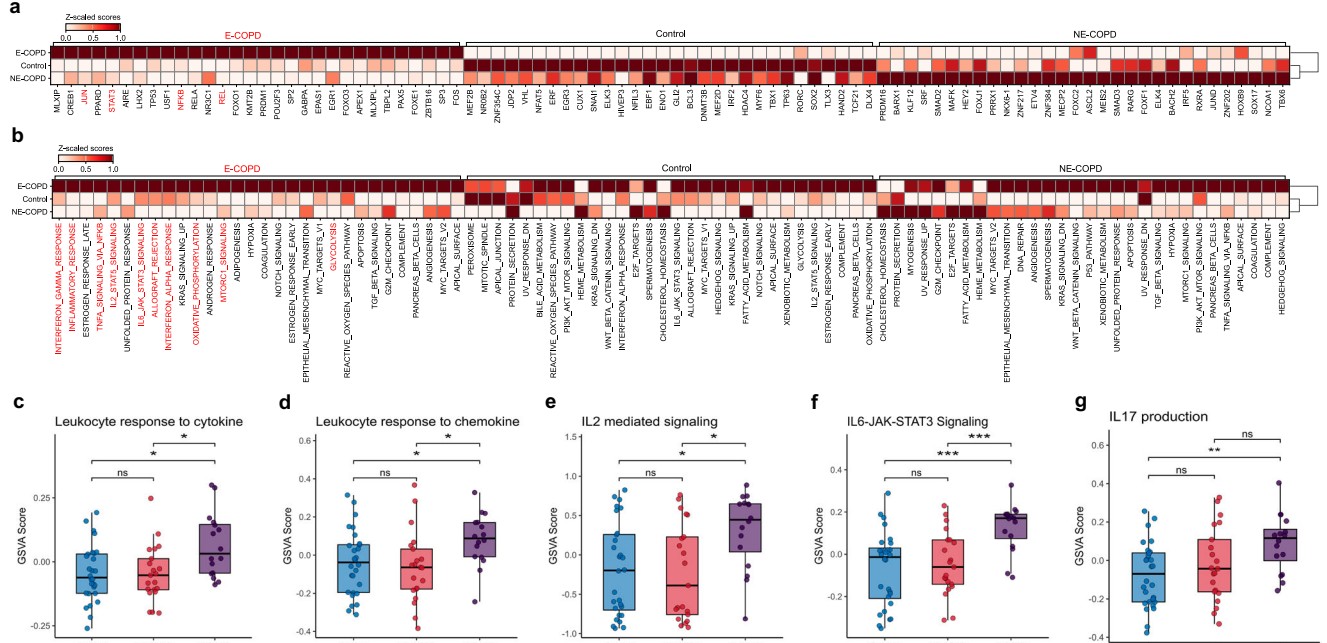

**Fig. 2 | Unique transcriptional signatures of CD4 T cells in Emphysematous COPD (E-COPD). a** Transcription factor activity estimation with a univariate linear model of CD4 T cells using decoupleR in Controls, Nonemphysematous COPD (NE-COPD), and Emphysematous COPD (E-COPD) in the in-house discovery dataset. **b** Functional enrichment of biological terms by over-representation analyses (ORA) of CD4 T cells across Controls, NE-COPD, and E-COPD using Hallmark pathway gene set in the in-house discovery dataset. Gene set variation analyses (GSVA) of Leukocyte response to cytokine (Gene Ontology) (**c**), Leukocyte response to chemokine (Gene Ontology) (**d**), IL2 mediated signaling (KEGG) (**e**), and IL6-JAK-STAT3 Signaling (KEGG) (**f**) and IL17 production (**g**) across three disease groups in the in-house discovery dataset. Statistical significance between the 3 groups was determined using the Kruskal–Wallis' test. Pairwise comparisons were performed using Wilcoxon rank-sum test with Holm's correction. Significance code: $p < 0.05 (*)$, $p < 0.01 (**)$, $p < 0.001 (***)$, ns: non-significant.

E-COPD but was increased in NE-COPD relative to control (Fig. 4a). Notably, alterations in the relative abundance of the NR3C1 subset were independent of individual patients' glucocorticoid therapy and the types of steroids used (Fig. 4b, c). Mapping of NR3C1 targets in Omnipath[51] showed that pro-inflammatory IL6, RELA, and JUN are among the inhibited targets, indicating the possible role of this subset of T cells in inflammation control (Fig. 4d). Mapping the signature genes of the NR3C1 CD4 subset to pathway gene sets showed that TGFβ signaling is among the top upregulated pathways, suggesting that TGFβ signaling might be critical for either the function or maintenance of this T cell subset (Fig. 4e).

Because CD4 T cells exhibited heightened responses to chemokines and cytokines in E-COPD (Fig. 2c–fd), we next performed interactome modeling to identify cellular interactions between different immune cells in this cohort. Interactome modeling estimated that CXCL signals mediate interactions between myeloid and T cells (Supplementary Fig. 5a–c). CXCL signaling activities were largely absent between NR3C1+ CD4 T cells and myeloid in either the control or NE-COPD groups (Fig. 4f, g, Supplementary Fig. 5a, b). In contrast, interactions between myeloid cells, especially pulmonary macrophages and NR3C1+ CD4 T cell subset via CXCL signaling, were strongly predicted in E-COPD (Fig. 4h, Supplementary Fig. 5c). CCL signals were also predicted to mediate interactions between T cell subsets and myeloid cells in all three groups (Supplementary Fig. 5d–f). However, in E-COPD, CCL signals were primarily predicted to mediate interactions between T cell subsets (Supplementary Fig. 5f), whereas in NE-COPD and controls, CCL signals were largely predicted to mediate myeloid-T cell interaction (Supplementary Fig. 5d, e).

Together, these findings suggest that, independent of steroid usage, reduced relative abundance of the NR3C1+ subset of CD4 T cells in the E-COPD cohort might be a distinguishing factor that separates them from the NE-COPD and controls. These findings also suggest a specific potential interaction between lung myeloid compartments, including macrophages with NR3C1+ CD4 T cells, that may contribute to their reduced relative abundance in E-COPD.

## Divergent abundance of lung PPARG+ macrophages associates with disease phenotypes

Because we found a potential interaction between lung macrophages and the NR3C1+ T-cell subset, we next examined the heterogeneity of human pulmonary macrophages in the lungs of the same cohort. We classified human lung macrophages into four distinct subsets: (1) a proliferative subset marked by the expression of MKI67, (2) PPARG macrophages, (3) Monocytic macrophages expressing CD14 and IL1β, and (4) C1Q macrophages (Fig. 5a, b). These four subsets of macrophages exhibited distinct gene expression and transcription factor activity profiles (Fig. 5c, d). All subsets of pulmonary macrophages, including PPARG, monocytic, and C1Q pulmonary macrophages in E-COPD, showed upregulated inflammatory pathways such as IFN-γ signaling (Supplementary Fig. 6a–c), which has been associated with the inhibition of glucocorticoid signaling[52]. In NE-COPD, PPARγ macrophages were relatively increased (Fig. 5e) suggesting increased renewal or persistence of this subset of macrophages. In contrast, the decreased relative abundance of PPARG macrophages in E-COPD mirrored the reduced proportion of NR3C1+ CD4 T cells in E-COPD (Figs. 4a & 5e). Further, in NE-COPD, where PPARγ macrophages were relatively increased, pairwise comparisons of ligand-receptor activities predicted increased interactions between PPARγ macrophages with NR3C1+ CD4 T cells through integrins, cell adhesion molecules and chemokine/cytokine receptors such as ITGB2, ALCAM, and CXCR4 (Supplementary Fig. 7a–c). In E-COPD, increased interaction through IL-1β and ADRB2 was estimated to be upregulated between PPARγ and NR3C1+ CD4 T cells (Supplementary Fig. 7b) compared to NE-COPD. Notably, however, in NE-COPD, interactions through SIGLEC1-SPN(CD43) between PPARγ macrophages and NR3C1+ CD4 T cells were estimated to be increased (Supplementary Fig. 7e). CD43 is critical for CD4 T cell trafficking[53] and is a known counter-receptor for SIGLEC1[54].

## NR3C1+ CD4 subset correlates with lung inflammation

Given the limitation of scRNA seq studies in identifying tissue-based associations between immune cells, we next performed spatial

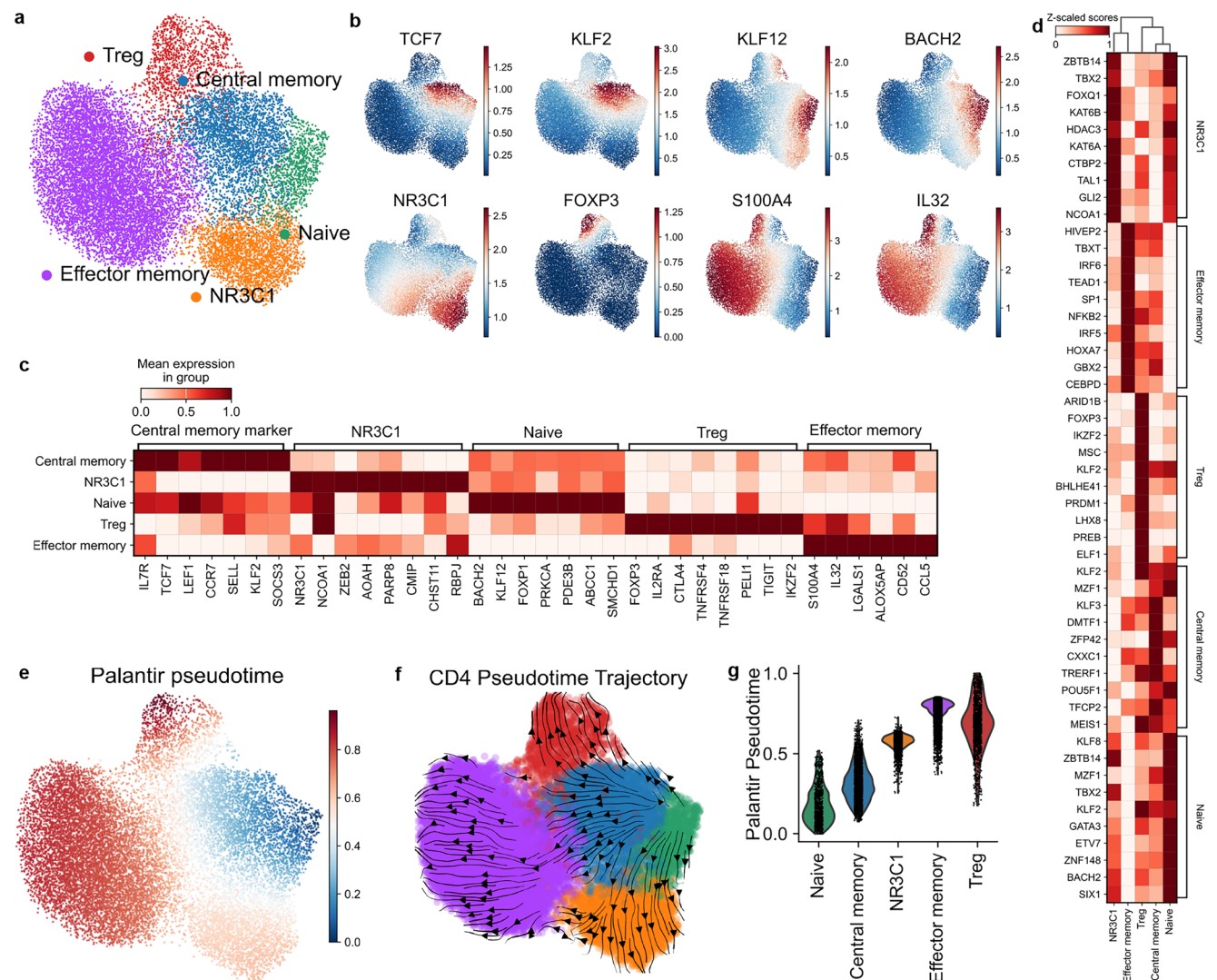

**Fig. 3 | Identified CD4 subsets in the human lungs. a** UMAP embedding of identified CD4 subsets in the in-house discovery dataset. **b** UMAP embedding of major subset marker expression: Central memory CD4 (TCF7, KLF2), Naïve CD4 (KLF12, BACH2), NR3C1 (NR3C1), Treg (FOXP3), Effector memory (S100A4, IL32). **c** Heatmap of gene signature expression in the identified 5 subsets of CD4 T cells in the in-house discovery dataset. **d** Transcription factor activity was estimated with the univariate linear model using decoupleR for each CD4 subset. **e** UMAP embedding of Palantir pseudotime. **f** Differentiation trajectory of CD4 subsets estimated by cellrank using Palantir pseudotime. **g** Violin plots of palantir pseudotime values for identified CD4 subsets.

transcriptomics to establish the correlation between NR3C1⁺ CD4 T cells and inflammatory cells in the lungs. We employed the Nanostring GeoMx platform to measure bulk RNA transcript levels in microscopic regions of interest (ROI), including the lung parenchyma, using samples from the same cohorts as the in-house single-cell RNA sequencing dataset. This spatial transcriptomics data of the lung parenchyma from the GeoMx platform were then deconvoluted for the relative abundance of cell types with CibersortX using the in-house single-cell RNA-sequencing defined cell type signatures as references.[55] Our spatial transcriptomic data showed that within the lung parenchyma ROIs, the relative abundance of NR3C1⁺ CD4 T cells positively correlates with neutrophils and migratory dendritic cells in NE-COPD (Fig. 6a). This is consistent with our scRNA-seq data where we found an increase in the relative abundance of NR3C1⁺ CD4 T cells in NE-COPD.

Next, we correlated the relative abundance of CD4 subsets using the single-cell RNA-sequencing dataset and the plasma chemokine/cytokine levels in the same E-COPD and NE-COPD cohorts. We found that in E-COPD cohort, CXCL13, CXCL5, and CXCL1 positively correlated with the relative abundance of NR3C1⁺ CD4 T (Fig. 6b). CXCL13 is a major chemokine for germinal center formation[56], whereas CXCL5[57] and CXCL1[58]

are involved in neutrophil chemotaxis. These findings suggest that the presence of NR3C1⁺ CD4 T cells in the lung is significantly associated with neutrophils in the regions, and the abundance of this T cell subset is associated with chemokines that are critical in the lymphoid follicle development.

## The CXCR6^high subset of CD4 effector memory correlates with preserved lung function in emphysema

We next examined whether the relative abundance of any CD4 subset correlates with disease severity (i.e., %LAA) or lung function measurement in each cohort. We found that the CD4 T cell effector memory subset positively correlated with predicted lung function as measured by forced expiratory volume (FEV1) percent in the E-COPD group but not in NE-COPD or controls (Supplementary Fig. 8a), and no significant correlations have been observed in other CD4 subsets (Supplementary Fig. 8b–f). Because a higher relative abundance of CD4 T cell effector memory was uniquely linked to preserved lung function in E-COPD, we next explored the heterogeneity within this population. We found that CXCR6, a receptor for CXCL16 and IL15, separates the effector memory into CXCR6^high and CXCR6^low subpopulations, each showing unique signatures (Fig. 7a–c). This CXCR6-based classification of effector memory CD4 could also be observed

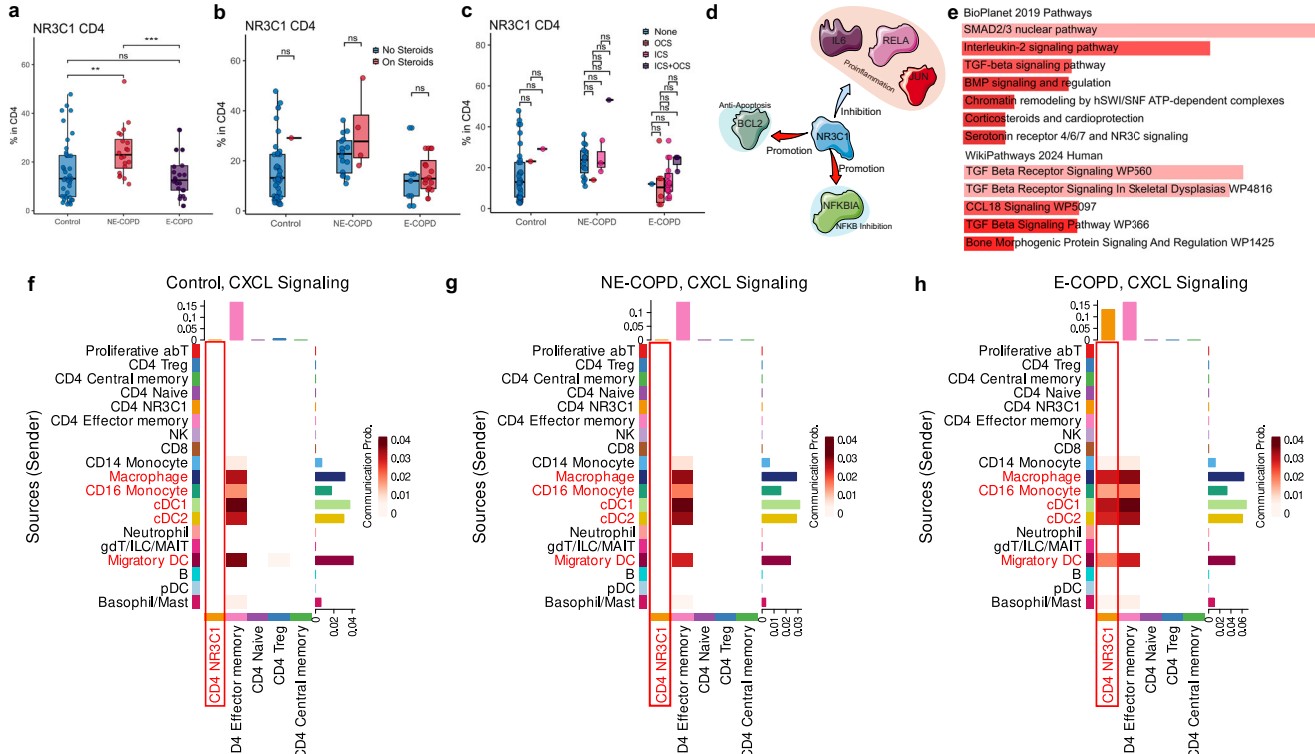

**Fig. 4 | Reduced NR3C1 CD4 T cells and increased myeloid cell-NR3C1 CD4 interactions in Emphysematous COPD (E-COPD) compared to none-mphysematous COPD (NE-COPD). a** Percentages of NR3C1 in CD4 T cells in the in-house discovery dataset. **b** Percentages of NR3C1 in CD4 T cells across 3 disease groups stratified by steroids usage. **c** Percentages of NR3C1 in CD4 T cells across 3 disease groups stratified by types of steroids used. **d** Top downstream targets of NR3C1 genes mapped using the Omnipath database, blue arrows indicate inhibition by NR3C1, and red arrows indicate induction by NR3C1. **e** EnrichR analyses results of NR3C1 CD4 subset signature genes using BioPlanet 2019 and WikiPathway 2019 Human gene sets. Interactions via CXCL signaling between identified major cell types and CD4 subsets in Control (**f**), NE-COPD (**g**), and E-COPD (**h**). Horizontal bar plots represent incoming signal strength estimated by Cellchat. A vertical bar plot represents outgoing signal strength. Tiles in the heatmap represent the summation of incoming and outgoing signal strengths. Statistical significance between the three groups was determined using the Kruskal–Wallis test. Pairwise comparisons were performed using the Wilcoxon rank-sum test with Holm's correction. Significance code: $p < 0.05$(*), $p < 0.01$(**), $p < 0.001$ (***). Artwork was generated from Bioicons (https://bioicons.com/) and NIH NIAID BioArt Source (https://bioart.niaid.nih.gov/). Modifications of the artwork were performed in Inkscape (https://inkscape.org/).

in the validation dataset, with CXCR6[High] and CXCR6[Low] subsets showing similar gene signature expressions and estimated differentiation trajectory (Supplementary Fig. 9a–g). CXCR6[high] CD4 effector memory expressed a higher level of CTLA4, an immune checkpoint molecule, compared to the CXCR6[low] subset (Fig. 7c). Further, CXCR6[low] effector memory showed higher KLF2 transcription factor activity (Fig. 7d), which has been associated with the downregulation of chemokine receptors and the potential establishment of tissue residence in mice[59,60]. Unsupervised pseudo-temporal modeling with CytoTRACE[61] suggested that CXCR6[low] represents a more differentiated state of CD4 effector memory (Fig. 7e, f). Consistently, in contrast to CXCR6[high], driver genes for CXCR6[low] CD4 effector memory differentiation were associated with the activation of CD4 T cell receptors (Fig. 7g). Notably, we found that the relative abundance of CXCR6[high] effector memory in CD4 T cells positively correlated with lung function in the E-COPD group (Fig. 7h), mirroring the overall effector memory CD4 correlation (Supplementary Fig. 8a) and no significant correlations were observed between CXCR6[Low] CD4 effector memory and lung function (Supplementary Fig. 8b).

Together, these results suggest that higher expressions of immune checkpoints in CXCR6[High] CD4 Effector memory might contribute to the preserved lung function in E-COPD.

### Increased interactions between migratory DC and CXCR6[High] CD4 effector memory via immune checkpoint molecules

Because CXCR6[high] CD4 effector memory in CD4 positively correlated with preserved lung function in E-COPD, and immune checkpoint molecules

(CTLA4, PDCD1, LAG3, and CD96) were among the signature genes expressed in this subset, we next performed interactome analyses to identify potential mechanisms for their function in E-COPD. Global interactome analyses suggested that migratory DC might be a major interaction partner with CXCR6[High] CD4 effector memory in all three groups (Supplementary Fig. 5a–c & Fig. 8a–c). In the E-COPD group, we found a shifting pattern between CXCR6[high] effector memory T cells and migratory DCs that exhibited increased interactions via CTLA4-CD80/CD86 (Fig. 8c). We next used pairwise comparisons of ligand-receptor activities to identify potential differentially regulated ligand-receptor pairs between migratory DCs and CXCR6[high] across three disease groups. Comparing NE-COPD to controls (Fig. 8d), we found that in NE-COPD, activities of TGFβ-TGFβR2, TGFβ-ITGB1, IL1β-IL12RB2, and IL15-IL2RB ligand-receptor pairs were down-regulated in CXCR6[High] CD4 subsets whereas activities of MIF-CD74, MIF-CD44, CCL5-CCR5, CCL4L2-CCR5, and CCL4-CCR5 ligand-receptor pairs were upregulated (Fig. 8d).

Comparing E-COPD to NE-COPD, we found that activities of MIF-CXCR4, MIF-CD74, MIF-CD44, and CXCL11-CCR5 ligand-receptor pairs were upregulated in E-COPD, whereas IL7-IL2RG, IL1β-IL12RB2, IL18-IL18RAP and ICAM1-IL2RG ligand-receptor pairs were upregulated in NE-COPD (Fig. 8e). Notably, interaction via the immune checkpoint LAG3 through the LGALS3-LAG3 ligand-receptor pair was upregulated in E-COPD compared to NE-COPD. When comparing E-COPD to controls, we found that CXCL10-mediated interactions (e.g., SDC4, DPP4, CXCR3), and ligand-receptor pairs associated with immune checkpoint molecules (CD48-PDCD1, PVR-CD96, CD274-PDCD1) were upregulated in

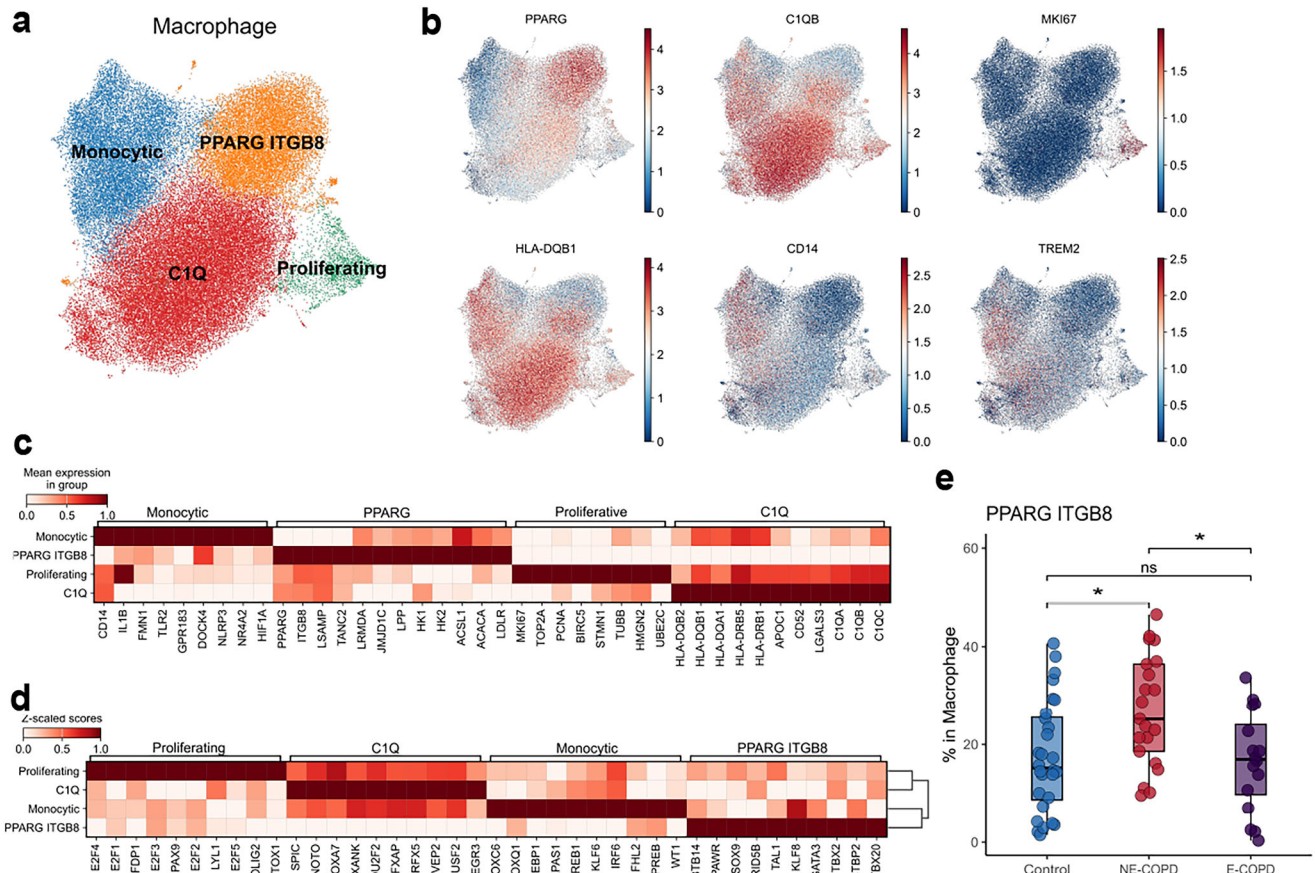

**Fig. 5 | Reduced tissue resident macrophages in emphysematous COPD (E-COPD) compared to nonemphysematous COPD (NE-COPD). a** UMAP embedding of identified pulmonary macrophage subsets. **b** UMAP embedding of major lineage markers' expressions for identified macrophage subsets. PPARG ITGB8 macrophage (PPARG), C1Q macrophages (C1QB, HLA-DQB1), Monocytic macrophages (CD14, TREM2) and proliferative macrophages (MKI67). **c** Heatmap of gene signature expression of identified macrophage subsets. **d** Heatmap of

DecoupleR estimation of transcription factor activity in identified macrophage subsets. **e** Percentage of PPARG ITGB8 macrophages in pulmonary macrophages. Statistical significance between the 3 groups was determined using the Kruskal–Wallis test. Pairwise comparisons were performed using the Wilcoxon rank-sum test with Holm's correction. Significance code: $p < 0.05$(*), $p < 0.01$(**), $p < 0.001$ (***).

E-COPD (Fig. 8f). These findings were specific to E-COPD because the interactions via immune checkpoint molecules were not observed in NE-COPD or control (Fig. 8d).

Together, these findings suggest that while pro-inflammatory interactions were likely to be upregulated in both NE-COPD and E-COPD; however, CXCR6$^{High}$ CD4 effector memory upregulated interactions via immune checkpoint in E-COPD but not NE-COPD. These findings might be indicative of T cell exhaustion but also control of T cell activation through immune checkpoints contributing to a lower inflammatory state of CXCR6$^{High}$ CD4 effector memory may partially explain the preserved lung function in E-COPD. Consistently, CD4 CXCR6$^{high}$ effector memory display the highest expression of the immune checkpoint molecule CTLA-4 in E-COPD (Fig. 8g, h). Furthermore, in E-COPD, migratory DCs exhibit a signature consistent with heightened oxidative phosphorylation and DNA damage responses (Supplemental Fig. 10), which have been associated with defective DC migration and activation[62].

## Discussion

We performed scRNA transcriptome profiling of human lung tissue from a cohort of well-characterized patients with different stages of lung disease. Consortium datasets that include large sample sizes and profile pulmonary cellular landscapes, such as the human lung cell atlas (HLCA) project[63], often combine a multitude of lung diseases. The HLCA project combined 49 datasets of the human respiratory system (upper and lower airways) from 486 human subjects spanning over ten lung diseases. The HLCA has

provided a consensus of major cell lineages in the lungs but lacks granular examination of immune cell types and how they are altered in specific conditions. Here, we leveraged our large sample size and detailed disease phenotype information for each subject to identify how the immune system could be specifically altered in human emphysema.

The most informative approach to examining tissue cellular landscapes in smoke-induced lung diseases (e.g., COPD with or without emphysema variant) requires access to surgical lung resections combined with high-throughput transcriptomic analysis. We provide objective evidence that CD4 T cells in the lungs of E-COPD are distinct from NE-COPD and controls. Although smokers can present with heterogeneous respiratory insufficiency, emphysema often goes undetected in the early stages of the disease. Pulmonary function testing (PFT) identifies airflow obstruction and has been widely used to assess disease severity in smokers. However, emphysematous tissue destruction can be found in some smokers with chest computed tomography (CT) in the absence of any detectable PFT abnormalities[64,65]. Although such discordance between anatomical tissue destruction and pulmonary function is frequently seen in clinical settings[66], CT-based quantification of emphysema is rarely used in clinical or pre-clinical studies. Accurate phenotyping of COPD requires thorough diagnostic approaches that include not only physiological measurements of respiratory mechanics but also imaging studies that measure the degree of pulmonary parenchymal degeneration.

Several lines of evidence corroborate our current findings that αβ T lymphocytes are increased in the lungs of smokers with emphysema. First,

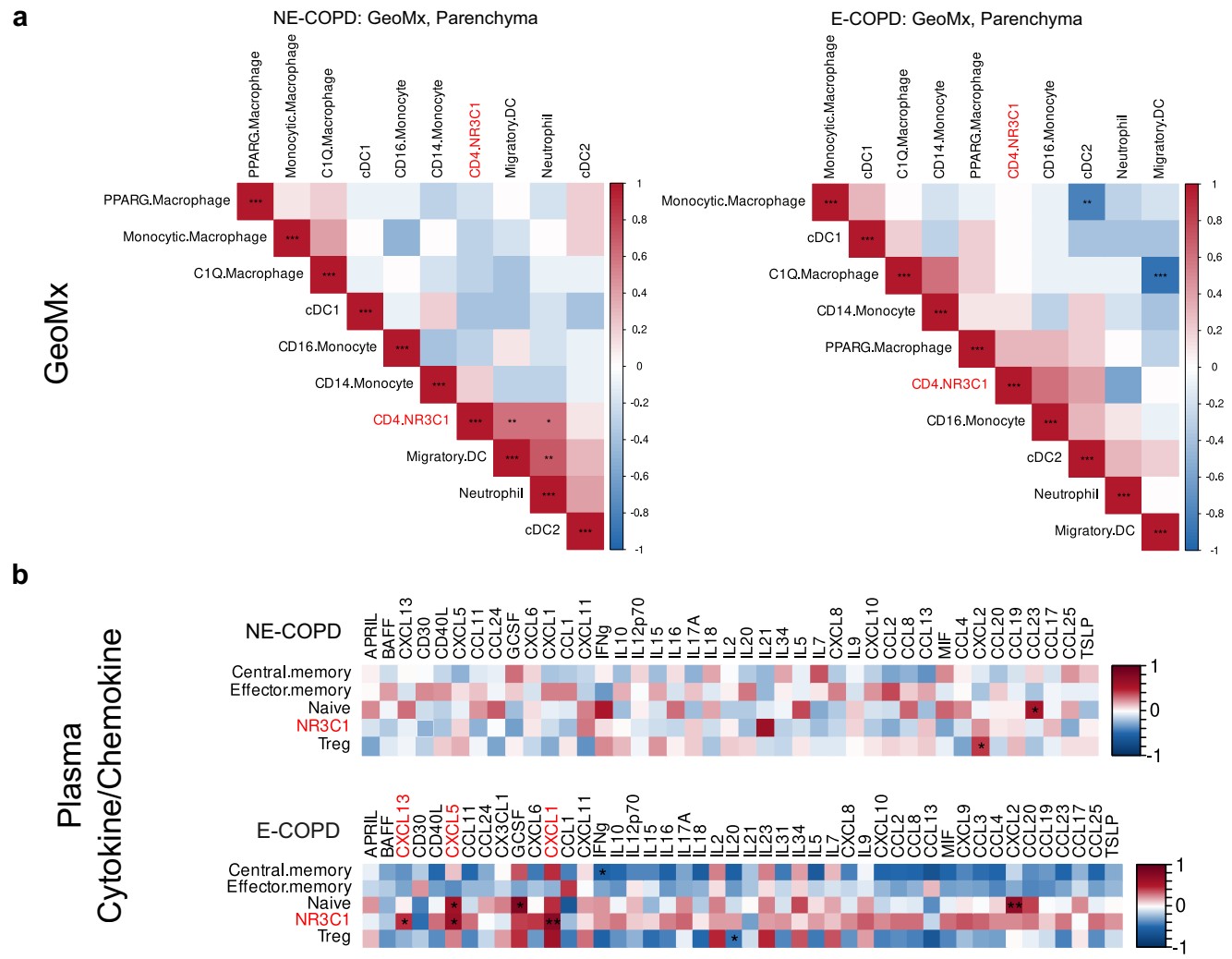

**Fig. 6 | Correlation of NR3C1 CD4 subset and inflammation. a** Spearman correlations of deconvoluted relative abundances were performed for NE-COPD and E-COPD groups. **b** Spearman correlation of the relative abundance for CD4 subsets and plasma cytokine/chemokine levels in NE-COPD and E-COPD. Significance code: *$p < 0.05$; **$p < 0.01$; ***$p < 0.001$.

the presence of aggregates of T and B lymphocytes in the lung parenchyma of smokers that correlate with disease severity indirectly supports the role of the adaptive immune system in disease progression[67,68]. IFN-γ and IL-17A expressing T helper cells have been shown to play a pathogenic role in smoke-induced emphysema in humans and animal models of emphysema[42]. The mechanism for IFN-γ- and IL-17-associated disease pathogenesis includes their downstream effector chemokines, such as IFN-γ-inducible protein of 10 kDa (IP-10 or CXCL10) and monokine-induced by IFN-γ (MIG or CXCL9) that upregulate matrix metalloproteinases (MMP)12 and MMP9[69]. These findings support a role for CD4 subsets (i.e., Th1 and Th17) in the induction of proteinases that cause lung parenchyma destruction in progressive emphysema.

By stratifying COPD with %LAA-defined emphysema in the COPD cohort, we isolated the differences between emphysematous and none-mphysematous COPD phenotypes. Globally, we discovered positive enrichment of αβ T lymphocytes and monocytes in E-COPD and positive enrichment of macrophages in NE-COPD. Gene sets enrichment analyses suggested that T cells in E-COPD exhibit distinct functional characteristics compared to NE-COPD, emphasizing the importance of stratifying COPD by %LAA-defined pulmonary emphysema in addition to airflow obstruction. We found that CD4 T cells in emphysema exhibited an activated transcriptomic profile indicative of heightened metabolism, augmented cytokine signaling, and enhanced inflammatory response.

Interactome inference suggested that CD4 effector memory might be sending more signals to various immune cells and could be independent of cDC in self-maintenance. In E-COPD, more interactions between different subsets of T cells via CCL signaling suggest that emphysema-associated CD4 T cells might be more autonomous and possess autoreactive properties. In addition, our findings suggest a decrease in NR3C1+ CD4 T cells in E-COPD, which was independent of the usage of corticosteroids. In silico, we also detected increased interactions between macrophages and NR3C1+ CD4 T cells via CXCL signaling in E-COPD but not in the other two groups. Expression of NR3C1 has been documented in both CD8[70] and CD4[71] T cells. Glucocorticoid responses in T cells not only suppress inflammation[70] but also upregulate IL-7 receptor expression[72] that promotes tissue accumulation of T cells in anti-microbial responses[71]. Monocytes and macrophages produce glucocorticoids, which are ligands for NR3C1 and have been shown to maintain the endogenous expression of NR3C1 in T cells[70]. In the present dataset, the relative reduction of NR3C1+ CD4 T cells in E-COPD mirrored the reduction of PPARG macrophages in E-COPD, suggesting a possible mutual regulation between these cell populations in the lungs. Indeed, interactome modeling suggested an emergence of NR3C1+ CD4 T cells and macrophages interactions via CXCL signaling in E-COPD, which was absent in controls and NE-COPD. Higher expression of NR3C1 in CD4 T cells has been positively correlated with expiratory reserve volume in obese pediatric asthma[73],

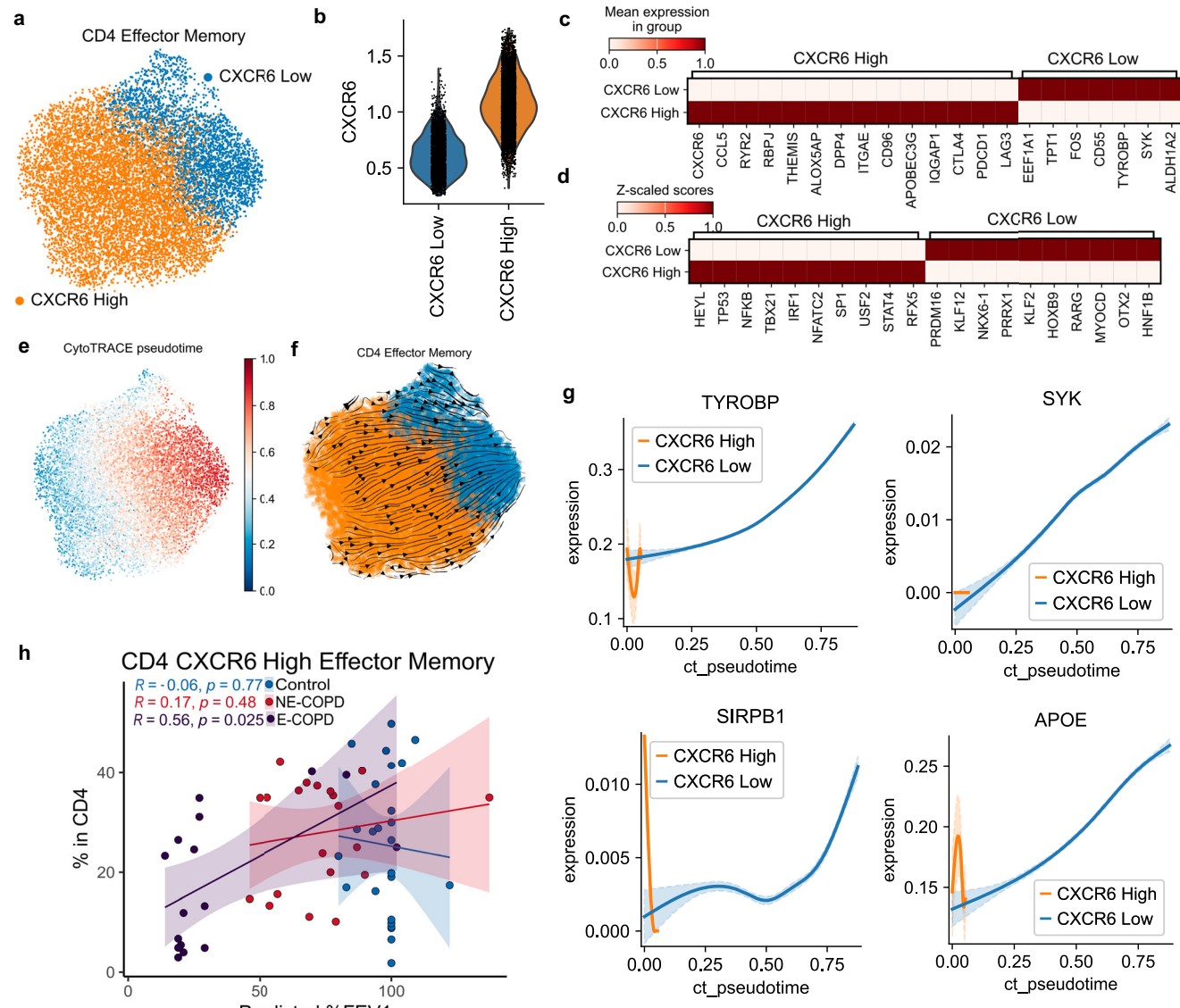

**Fig. 7 | CXCR6^High effector CD4 positively correlates with %FEV1 in E-COPD.**
**a** UMAP embedding of CD4 effector memory subsets in the in-house discovery dataset. **b** Violin plot of CXCR6 expression in CD4 effector memory subsets. **c** Heatmaps of gene signature expression and (**d**) decoupleR estimation of transcriptional factor activities in CXCR6^High and CXCR6^Low CD4 effector memory subsets. **e** UMAP embedding of CytoTRACE pseudotime in CD4 effector memory. **f** Cellrank estimation of differentiation trajectory in CD4 effector memory subsets.

Higher values of cytoTRACE pseudotime indicate a higher level of differentiation. **g** Driver genes for CXCR6^Low CD4 effector memory differentiation from CXCR6^High CD4 effector memory. Expressions of driver genes (y-axis) were plotted against CytoTRACE pseudotime (x-axis), the shadow of the curve denotes 95% confidence intervals. **h** Spearman correlation of CXCR6^High CD4 effector memory percentages in CD4 and %FEV1 in 3 disease groups.

suggesting their potential protective role in the lung. In NE-COPD, data analysis predicted increased interactions between PPARγ macrophages and NR3C1⁺ CD4 T cells through SIGLEC1-CD43. Prior studies have shown that SIGLEC1 interacts with CD43 in CD4 T cells, thereby suppressing pro-inflammatory cytokine production[74]. Together, increased interaction via CD43 might explain the parallel increase of PPARγ macrophages and NR3C1⁺ CD4 T cells in NE-COPD and suggest that these macrophages could promote the trafficking/expansion of NR3C1⁺ CD4 T cells in the lung. However, whether NR3C1⁺ CD4 T cells are protective or pathogenic in human emphysema and COPD merits further investigation in experimental and clinical studies (**Graphic Abstract**).

Increased airflow obstruction with escalated pulmonary tissue destruction predicts the worst prognosis in COPD[75]. In the present study, we found that the relative abundance of CXCR6^High effector memory CD4 T cells positively correlates with a higher predicted %FEV1 in the emphysema cohort. CXCR6^High CD4 effector memory represents the recent lymph

node emigrants and exhibits a lower level of differentiation compared to CXCR6^Low CD4 effector memory in the lung. We also found that CTLA-4 was expressed at a higher level in CXCR6^High than in CXCR6^Low CD4 effectors. In E-COPD, migratory dendritic cells showed the potential to interact more with CXCR6^High through CD80/CD86-CTLA4 signaling. These interactions likely result in checkpoint-mediated immune suppression, providing a new mechanistic insight into the increased CXCR6^High association with preserved lung function in emphysema. CXCR6^High CD4 effector memory also expresses higher levels of IFN-γ. Paradoxically, IFN-γ-expressing CD4 T cells are protective in acute traumatic injury-associated inflammation in the central nervous system (CNS), most likely through the regulation of myeloid cells in the CNS[76]. Whether such observations are translatable in pulmonary emphysema-associated inflammation requires further investigations. Furthermore, driver genes of CXCR6^Low CD4 effectors map to pathways associated with activating T cell receptor signaling. These observations suggest that CXCR6^Low CD4 effectors might be the

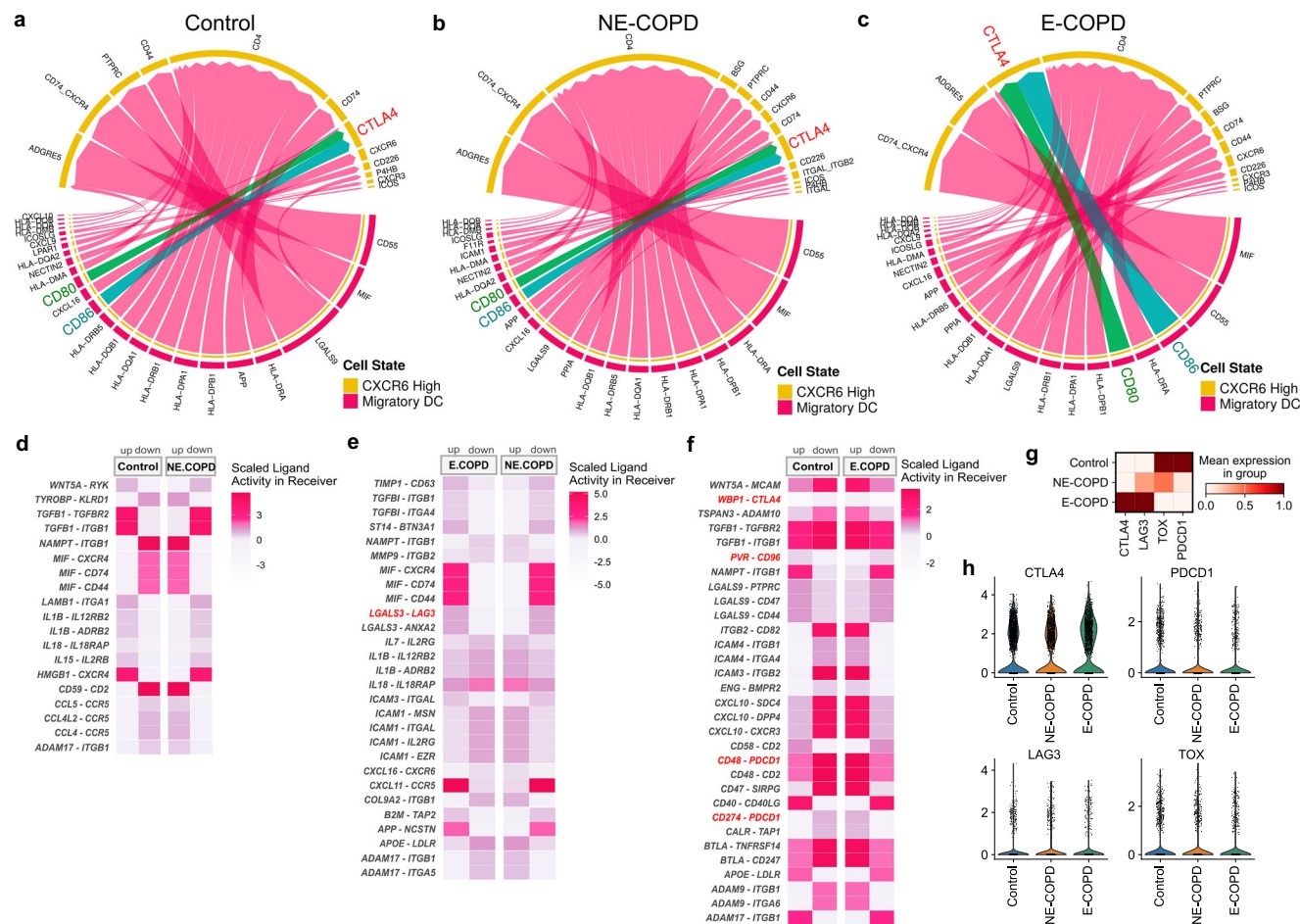

**Fig. 8 | CXCR6$^{High}$ CD4 T cells exhibit increased interactions with migratory DCs via immune checkpoints. a–c** Interaction patterns between migratory DC and CXCR6$^{High}$ CD4 effector memory estimated by CellChat. Chord plots depict signals sent from migratory dendritic cells and received by CXCR6$^{High}$ CD4 effector memory. The width of the chord indicates the strength of the interaction via the indicated ligand-receptor pairs. Interactions with CTLA4 were highlighted in blue (CD86) and green (CD80). **d–f** Pairwise comparison results of differentially regulated ligand-receptor pairs estimated by Multinichenetr between migratory DC and CXCR6$^{High}$ CD4 effector memory. **g, h** Expression of CTLA4, LAG3, TOX and PDCD1 across 3 groups in the in-house discovery dataset.

culprit in propagating the inflammation in E-COPD and the consequent decline of pulmonary function.

Limitations of the study include the lack of in vivo data that could validate the predicted cell differentiation trajectories using lineage tracing transgenic animal models. While these studies are beyond the scope of current study, our findings provide a strong rationale for opening new investigations using animal models of cigarette-smoke-induced emphysema. Although we used the NicheNet[77] algorithm to estimate and predict communications between cells in a pairwise fashion across COPD groups with and without emphysema, these findings should be followed up with human lung tissues collected from well-characterized smokers in future studies.

Future studies should opt for multi-omics approaches (protein, mRNA, DNA, and chromosomal accessibility) of single-cell sequencing. There is also a strong interest in identifying early immune activation markers in emphysema. Notably, a recent publication showed that CD8 T cells were increased in the early stages of COPD[31]. As such, future longitudinal cohort studies that include PFTs and emphysema quantification should be used to find immune signatures that could predict emphysema development. Moreover, with existing large-scale genetic studies of emphysema, single-cell DNA-level profiling will enable GWAS to assess the risk of emphysema development in smokers at unprecedented comprehensiveness and resolution. Indeed, large-scale GWAS-based discovery of single-nucleotide polymorphisms (SNPs) has identified three new loci (SOWAHB, TRAPPC9, and KIAA1462) associated with emphysema[78]. Future studies

should also include clonotype analyses of T cell receptors to examine in depth the expansion and evolution of T cell clonotypes and the antigen responsible for such expansion during emphysema pathogenesis.

## Materials and methods
### In-house dataset one
Control lungs were acquired from medically necessary surgical explants. COPD lungs and plasma were obtained from lung volume reduction or transplant surgeries as we have described[42]. Fresh human lung tissues and plasma were processed within 1 h post surgery. Measurement of Low Attenuation Areas below 950 Hounsfield Units threshold (%LAA) by chest computed tomography was performed as previously described[68]. Post-surgical lung tissues were assessed by a pathologist to identify cancer margins, and tissues were resected at least 10 cm away from macroscopically distinguishable malignant tissue if cancer was present. Lung tissues were cut into small pieces and then transferred to 10 mL RMPI 1640 media containing 0.5 mg/mL collagenase P (SKU11213857001, Millipore Sigma, USA) and 0.1 mg/mL of DNase I (SKU11284932001, Millipore Sigma, USA) before mechanical dissociation with gentleMACS (Miltenyi Biotech, USA) "m_lung_01_02" program. After completion of the "m_lung_01_02" program, tissue homogenates were incubated on a shaker at 37 °C for 30 min. Tissues were then mechanically dissociated with the gentleMACS (Miltenyi Biotech, USA) "Multi_B_01" program. Following completion of the program, single-cell suspension was passed through first a 100 mm cell

strainer and then a 40 mm cell strainer. Red blood cells were lysed using 10 mL 1X ACK Lysing Buffer (Catalog# A1049201, ThermoFisher, USA). Cells were then cryopreserved in freezing media (50% complete RPMI (Corning, USA) with 40% heat-inactivated FBS and 10% DMSO (Millipore Sigma, USA)) until single-cell library preparations.

## Study cohort and sample preparations

*CT quantification of human emphysema*. Emphysema characterization and definition have been published previously[79]. Briefly, clinical CT scans were assessed using the Chest Imaging Platform software. Emphysema was quantified using the percentage of low-attenuation areas below the −950 Hounsfield units (HU) threshold (%LAA-$_{950}$) in the specific lobes that were used for spatial transcriptomic analyses. Emphysema was defined as %LAA-$_{950}$ > 5%[32]. Patients were classified as Control (No Emphysema), COPD with No Emphysema, and COPD with Emphysema.

## Lung spatial transcriptomics and cell type deconvolution using CIBERSORTx

Transcript abundances of microscopic regions of interest on Formalin-fixed paraffin-embedded (FFPE) lung sections were measured using NanoString Digital Spatial Transcriptomics GeoMx technology according to the manufacturer's instructions. Cell deconvolution of GeoMx bulk gene expression data was performed with CIBERSORTx[80] (https://cibersortx.stanford.edu/) as previously described[79,81]. Briefly, count data from 50 replicates of 34 cell types in the in-house single cell RNA sequencing dataset from NE-COPD and E-COPD samples were uploaded to CIBERSORTx to construct a custom cell type signature matrix. Cell replicates were chosen only from patients used with both single-cell RNA sequencing data and GeoMx transcriptomic data. Default values were used for parameters such as G.min, G.max, q.value, filter, k.max, and sampling. The fraction option was set to *0.50*. The fractions module of CIBERSORTx was subsequently used to determine ROI cell proportions using normalized and batch-corrected GeoMx ROI expression data along with the previously constructed custom cell type signature matrix. The number of permutations was set to 100, with quantile normalization set to TRUE. Batch correction was incorporated using the S-mode option to correct for platform differences between the custom signature matrix constructed from the in-house single-cell RNA sequencing dataset and the GeoMx bulk RNA sequencing.

## LegendPlex multiplex cytokine assays

Peripheral blood was collected into EDTA-coated BD Vacutainers (Becton, Dickinson and Company, USA). Plasma samples were acquired by centrifugation of whole blood at $2000 \times g$ for 15 min at room temperature and cryopreserved for multiplexed cytokine measurement. Multiplex cytokine assays were performed at UT Southwestern (UTSW) Microarray Core Facility using the ProcartaPlex™ Human Immune Response Panel, 80plex panel (Catalog EPX800-10080-901, Invitrogen, MA, USA) and detected using the Luminex xMAP technology platform according to the manufacturer's instructions.

## Generation of single-cell RNA sequencing data

*In-house dataset one*. Cryopreserved single-cell suspensions from the lung were thawed at 37 °C water baths and resuspended in 0.04% BSA containing 1X PBS. Viability was determined by Trypan Blue exclusion assays. Single-cell libraries were prepared using Chromium Next GEM Single Cell 5′ Reagent Kits v2, R2-only (10X Genomics, USA). Libraries were sequenced using the NovaSeq6000 platform (Illumina, USA). Libraries were prepared and sequenced at the University of Southern California Sequencing Core facility. Raw sequencing Fastq files were mapped to the human reference genome GRCh38-2020-A (compiled by 10XGenomics) and count matrixes were generated using the Cell Ranger v8.0 software (10XGenomics, USA).

*Validation dataset two*. Detailed patient information can be found in the original publication[34]. Control lung tissue were obtained from rejected donor lungs. Emphysematous lung samples were from explants of end-stage COPD patients who underwent lung transplantation. Cryopreserved single-cell suspensions were used for single-cell RNA sequencing library preparation. H5 format count matrices were downloaded from the National Center of Biotechnology Information (NCBI) Gene Expression Omnibus (GEO) database (GEO Accession: GSE136831).

## Pre-processing and quality control of single-cell data

Count matrices from both datasets were preprocessed using the SCANPY package version 1.11.1[82]. Cells with fewer than 500 genes and genes that were present in less than 10 cells were removed. Further quality control was performed by automatic thresholding using the Median absolute deviation[83]. Cells were considered outliers if they differed by 4 median absolute deviations in log1p_total_counts, log1p_n_genes_by_counts, pct_counts_in_top_20_genes, and pct_counts_mt. The mitochondrial transcript count percentage was further filtered at less than the 10% threshold[84]. Cells with high-complexity transcriptomes (>6500 genes) were removed[47]. Nuisance transcripts (mitochondrial, ribosomal, long noncoding transcripts) were removed before doublet estimation and doublet removal[85].

## Doublet removal

Doublet probability was estimated for each sample individually using DoubletDetection Version 4.2[85]. Estimated doublets were removed before the concatenation of all samples for each dataset.

## Batch-effect correction

Count matrixes are normalized and natural log-transformed using SCANPY's built-in functions pp.normalize_total and pp.log1p. Before data integration, highly variable genes were determined for each batch in each dataset using the following 3 thresholds: min_mean=0.0125, max_mean=3, and min_disp=0.5. Batch correction and integration were performed with Harmony[86].

## Cell-type identification markers

Cell type prediction was first performed with decoupleR Python implementation version 1.9.1[87] using the PanglaoDB cell marker database[88]. Predicted cell types are further examined with a manually curated canonical cell type marker dictionary (Table 3).

## Compositional and differential abundance analyses

Cell type enrichment was determined by scCODA compositional analysis[33]. scCODA was implemented through Pertpy (https://github.com/theislab/pertpy). For compositional analyses, proportions of each cell type for each sample were determined using scCODA and plotted using the pertpy.pl.coda.boxplots function. Collective proportions for each cell type between conditions (COPD with or without emphysema vs control) were plotted using the pertpy.pl.coda.stacked_barplot function. The reference cell type was selected based on abundance and dispersion in relative abundance. Abundance and dispersion were calculated and plotted using the pertpy.pl.coda.rel_abundance_dispersion_plot function for the identification of reference cell type. The cell type with the lowest total dispersion and presence passing the 0.9 threshold was selected as the reference cell type. The false discovery rate threshold for scCODA was set at 0.25.

## Subset determination and re-clustering of T cells and macrophages

Following canonical cell type annotation, major cell types of interest were further extracted. Continuous covariates such as 'pct_counts_mt', 'total_counts', and 'pct_counts_ribo' that were generated during quality control preprocessing were regressed out using the scanpy.pp.regress_out function. UMAP coordinates and Leiden clusters were then recalculated for clusters of interest, respectively. Leiden clusters were determined at 1 resolution. Marker genes for each cluster are then determined using Scanpy's tl.rank_genes_groups function with the t-test_overestim_var method.

**Table 3 | Canonical markers for cell type identification**

| Cell type | Canonical markers |
|---|---|
| cDC | BATF3 |
| cDC1 | CLEC9A, CADM1 |
| cDC2 | CST3, COTL1, LYZ, CLEC10A, FCER1A, DMXL2 |
| pDC | CLEC4C, IL3RA, GZMB, COBLL1, TCF4 |
| B cells | CD19, MS4A1, BANK1, CD79A |
| Plasma cells | IGHM, IGHD, IGHG1, IGHG2, IGHG3, IGHG4, IGHA1, IGHA2 |
| NK | GNLY, NKG7, CD247, GRIK4, FCER1G, TYROBP, KLRG1, FCGR3A |
| γδT | TRDV2, TRGV9, TRGV10 |
| MAIT | SLC4A10 |
| ILC | CD7, IL7R, ID2, PLCG2, GNLY, SYNE1 |
| αβT | CD3D, CD3E, CD3G, TRAC, TRBC1, LCK, FYN |
| CD4 αβT | CD4, CD40LG |
| CD8 αβT | CD8A, CD8B |
| Alveolar epithelia | SFTPC, SFTPB, EPCAM, AGER |
| Airway epithelia | SCGB1A1, MUC4 |
| Lung endothelia | PECAM1, LYVE1 |
| Fibroblast | COL5A1, COL3A1, COL18A1, MPG |
| Mesothelia | ITLN1, LRRN4, UPK38, MSLN |
| CD14 monocytes | FCN1, CD14 |
| CD16 monocytes | TCF7L2, FCGR3A, LYN |
| Monocyte | VCAN, CCR2, GPR183, CD14, CX3CR1, FCN1, CSF1R |
| Macrophage | MSR1, MARCO, FBP1, APOE, FABP4, LYZ, PPARG, MRC, LTA4H, CTSD, CTSL, FCGR3A |
| Neutrophil | AQP9, FUT4, FCGR3B, CXCR2, IL1R2, CD177, MMP9, CSF3R, S100A8, S100A9, CST3 |
| Proliferation | MKI67 |
| Basophil/Mast cell | CPA3, TPSAB1 |
| Megakaryocyte | PPBP |

## Pathway enrichment analysis

Pathway analyses were performed using gene set enrichment analyses (GSEA)[89,90] and gene set variation analysis (GSVA)[91]. Hallmark, Gene Ontology Biological Process (GOBP), and Kyoto Encyclopedia of Genes and Genomes (KEGG) gene sets from MSigDB[92] were used for mapping differentially expressed genes. GSEA was implemented through the Python-based decoupleR method.

## Interactome prediction using MultiNicheNet and CellChat

CellChat v2 was used to identify cellular interactions for each cohort, and the algorithm was run using default parameters[93]. MultiNicheNet was used for the pairwise comparisons of estimated cellular interactions[94]. Cell-cell communications were estimated in a pairwise fashion across disease groups (e.g., E-COPD vs control, E-COPD vs NE-COPD, NE-COPD vs control) using differentially expressed genes with a minimal logFC_threshold of 0.05, the maximum adjusted false discovery rate of 0.25, and a minimum fraction of 5% expression. The top 250 targets per ligand were considered for ligand activity analysis. Ligand-receptor interactions were prioritized based on three parameters using recommended configurations (https://github.com/saeyslab/multinichenetr/blob/main/vignettes/pairwise_analysis_MISC.md): (1) differential ligand-receptor expression, (2) Nichenet[77] estimation of ligand-receptor activity, and (3) cellular fraction of ligand-receptor expression.

## Statistical analyses

For statistical comparisons of continuous variables between more than two independent samples, a one-way analysis of variance was performed with the nonparametric Kruskal–Wallis's rank sum test. For comparisons between two independent samples, null hypotheses were tested using the nonparametric Wilcoxon rank sum test. Pearson's Chi-squared test was used for the test of independence of categorical variables. In cases where the sample size count was less than 5, Fisher's exact test was used instead of Pearson's Chi-squared test to compare 2 or more proportions. The statistical significance threshold for the $p$-value was set at 0.05.

## Reporting summary

Further information on research design is available in the Nature Portfolio Reporting Summary linked to this article.

## Data availability

Raw fastq sequences are uploaded to NCBI Sequence Read Archive (SRA) and can be accessed with the BioProject accession: PRJNA1282758. Cell-ranger processed count matrices are uploaded to NCBI Gene Expression Omnibus (GEO) and can be accessed with the GEO accession: GSE302339. GeoMx datasets used in this study can be accessed with GEO accessions GSE237120 and GSE292993. Code files used for the analysis can be accessed through Zenodo DOI 10.5281/zenodo.16341197.

## Code availability

Raw fastq sequences are uploaded to NCBI Sequence Read Archive (SRA) and can be accessed with the BioProject accession: PRJNA1282758. Cell-ranger processed count matrices are uploaded to NCBI Gene Expression Omnibus (GEO) and can be accessed with the GEO accession: GSE302339. GeoMx datasets used in this study can be accessed with GEO accessions GSE237120 and GSE292993. Code files used for the analysis can be accessed through Zenodo DOI 10.5281/zenodo.16341197.

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

## Acknowledgements

This work was supported by the US National Institutes of Health (NIH) grants R01 ES029442-01, R01 AI135803-01, and the US Veterans Administration (VA) Merit grant CX000104, DOD W81XWH-20-1-0607 to F.K.; HL117181, HL140398, R01 AI135803, and R41 AI124997 and VA Office of Research and Development grant I01BX004828 to D.C, DOD grant PR211314, R01HL155948 to MS, and NHLBI HL149744 to FP. Data analysis was performed on the HPC cluster at the Baylor College of Medicine BISR ATC, which includes equipment purchased using S10-OD032185 and systems administration supported in part by P30-CA125123 and Baylor College of Medicine. The authors have no competing interests to declare.

## Author contributions

F.K. and Y.Z. conceived the project, wrote the manuscript, and contributed to data analysis and interpretation. Y.Z. designed and conducted the data analyses, contributed to writing, and interpretation. S.P., L.S. provided technical assistance. D.C., F.P., R.S., S.O., J.M., and M.S. contributed to the writing, data analysis, and editing of the manuscript.

## Competing interests

The authors declare no competing interests.
