## [Transparent Peer Review file · Communications Biology]

Lung NR3C1+ and CXCR6high T cells distinguish immunopathogenesis of human emphysema

Corresponding Author: Professor Farrah Kheradmand

This manuscript has been previously submitted at another journal. This document only contains information relating to versions considered at Communications Biology.

Version 0:

Reviewer comments:

Reviewer #1

(Remarks to the Author)

General Comments:

In the manuscript entitled « Lung NR3C1+ and CXCR6high T cells distinguish immunopathogenesis of human emphysema », the authors propose to explore the specific alteration of the immune system in human emphysema, using single cell-RNA analysis performed on lung samples. The strengths of this study are the number of clinical samples (in total 62 tissue samples), their exhaustive clinical characterization (in particular radiographic quantification emphysema) as well as the careful and precise scRNA seq analysis. Notably, the authors also use a validation cohort (64 samples), from a publicly available scRNA seq dataset) to validate their findings. The main results include the identification of two subsets of immune cells associated with emphysema: NR3C1+CD4 T cells, reduced in emphysematous patients, and pro-inflammatory CD4 effector memory cells, increased in emphysematous patients.

Main weaknesses are the following:

In addition of the presence of emphysema and airflow obstruction, there are (too) many confounding factors in the three different groups (age, current/former smoker and treatment) that could strongly affect the results. Past and current smoking is especially problematic, as it is well known to affect immune responses.

Moreover, this study is very descriptive. Most of the « mechanistic » conclusions were drawn on pseudotemporal analysis and ligand-receptor analysis to predict cell differentiation trajectories and cell-cell interactions, respectively. Both types of analyses are based on tools that have serious limitations and bias. In the absence of any functional experiments, or at least spatial cellular analysis, the conclusions remain very speculative and do not bring much on the understanding of emphysema.

The authors focused on late stages of emphysema, which is another limitation, as it only shows the “final picture”. The inclusion of another group of patients, with emphysematous tissue destruction without major lung function alteration would have represented a significant advance to identify cellular alterations at early stages, to propose putative cellular culprits to initiate and propagate emphysematous lesions.

Major points:

#1. The control and COPD tissue samples did not come from the same type of sampling (medically necessary surgical explants for control tissues, lung volume reduction or transplant surgeries for COPD tissues). How did the authors control this potential major bias?

#2. The sc-analysis is performed on “lung tissue”. Given that emphysema is a specific alteration of alveoli, we would have expected a selection of the distal part of the lungs. Could the authors precise if such a selection was performed? If not, this should be included in the discussion as a limitation.

#3. There are many factors of variations between the three groups of patients: 1) the median age of the 3 different groups of patients is different: control patients are on average younger than the patients with COPD, and NE-COPD patients seem also older than E-COPD patients. 2) the percentage of current vs former smokers is also very different in the 3 groups. 3) the treatment (steroids) also differs between NE and E-COPD patients. How did the authors control these three potential major

biases?

#4. Quality control of single-cell data: The authors provided a total number of cells that passed quality control: “a total of 57,223 cells from the control, 55,632 cells from NE-COPD, and 46,847 cells from the E-COPD group passed quality control.” Could the authors provide the same quantifications per biological sample, to test for interindividual variations?

#5. The validation cohort is problematic: as the information regarding %LAA data was not available, there is only distinction between COPD and control patients. Then the validation of specific alterations in emphysematous patients identified with the original cohort is not possible.

#6. Figure 1e “epithelial cells showed negative enrichments in E-COPD compared to controls”. This seems to be due to a loss of airway epithelial cells, but not of alveolar epithelial cells. Can the authors comment on the fact that no alveolar epithelial loss was detected in the emphysematous condition in comparison to the other ones?

#7. Figure 2: “IL6 induces expression of IL21 and IL23R in CD4 T cells, which are upstream of RORC and IL17 expression, and STAT3 is also indispensable in the development of Th17 cells”: are Th17 cells identified in scRNA data, and how are they modified in the different groups of patients?

#8. The identification of the specific subset of CD4 T cells expressing glucocorticoid receptor NR3C1 and its reduction in E-COPD patients is interesting. The authors claimed that the “alterations in the relative abundance of the NR3C1 subset were independent of individual patient’s glucocorticoid therapy (Figure 4b)”. However, although it is not statistically significant, it seems that there is a higher level of NR3C1 positive cells in CD4+ cells from patients with steroids treatment. Could the authors comment on this point, and also provide a similar analysis than the one provided in Figure 4b, but with NR3C1 level of expression?

#9. Interactome modeling: based on transcriptional levels of ligand-receptor pairs, intercellular interactions are predicted, such as the interplay between pulmonary macrophages and NR3C1+ CD4 T cell subset in E-COPD patients. While this could be a good starting point, this is clearly not sufficient, in particular because transcriptional levels do not always reflect protein levels nor the proximity of the cells. The relevance of the interactions identified should be validated by quantifying protein levels and by spatially studying the relative distribution of implicated cells.

#10. P9 “Together, these results suggest that higher expressions of immune checkpoints in CXCR6High CD4 Effector memory might contribute to the preserved lung function in E-COPD.” This conclusion is very speculative, as it is based on correlations. The authors propose that the underlying mechanism is the interaction between CXCR6 high CD4 effector memory and migratory DC, based on interactome analysis. Can the authors provide data to support this hypothesis? Functional experiments are here required.

#11. Discussion “emphysematous tissue destruction can be found in some smokers with chest computed tomography (CT) in the absence of any detectable PFT abnormalities”: in fact, including these samples would be really interesting to look at early stages of emphysema development.

Reviewer #2

(Remarks to the Author)

In this paper, the authors applied single cell RNA sequencing on a large number of clinical samples, with detailed clinical information, and validated the findings in an online available database. The authors show that subsets of CD4 lymphocytes, expressing NR3C1 and CXCR, distinguish emphysema-COPD from COPD without emphysema. The validation cohort strengthens the findings of the authors. Furthermore, the authors provide detailed clinical data and link the single-cell RNA seq and interactome data to the clinical findings. Combined, the results provide detailed information and pinpoint potential targets for treatment of COPD-emphysema using elegant methods. There are some minor issues that can be addressed to improve the paper and strengthen the findings further.

In figure 1D, E there does not seem to be a difference in the AT1/AT2 cells between the groups, and in Figure 1D, there is also no difference between COPD and control in airway epithelia. The authors do mention that the number of cells derived from the individual groups was different. Was there a difference in the number of immune versus structural cells between the groups? In other words, was the emphysema-phenotype reflected on the number of immune/ structural cells? It may be informative to determine whether the LAA% correlates to any of the measured parameters, to determine whether emphysema severity is related to the subsets found.

On page 5, lines 2-4, the authors state that the CD8 T cell populations were enriched in E-COPD compared to NE-COPD, but not compared to controls. This seems contradictory, especially since several studies showed that CD8 T cells subsets are increased compared to controls. Please discuss these findings in relation to previously published findings showing that CD8 are increased in emphysema compared to control.

The expression of NR3C1 in the 2 cohorts seems different (Fig 3A, B versus Supplemental figure 3A, B). Whereas the CD4

subset in Fig 3A is very clearly separated from the other subsets (no/low expression of S110A4 and IL32), the cluster is not very distinguishable in the validation cohort (SFigure 3A,B). Please explain this difference. Furthermore, it would be helpful to depict also the source of cells (sample code) between the 3 (for the study cohort, control NE-COPD or E-COPD) or 2 (for the validation cohort) in figure S3A, Fig 3A, but also in Fig 5A, 6A, and S8A.

The authors state that CD4 T cells expressing the glucocorticoid receptor NR3C1 are enriched in NE-COPD, but reduced in E-COPD, which is one of the main messages of this paper. However, as shown in SFigure 1G, there was a significant difference in the steroid use between these groups, which may modulate expression of the receptor of interest (or localization of the receptor). Also gender and smoking history seems different between the groups. Although authors state that glucocorticoid use did not affect the outcomes, please provide some more information on the types of steroids used. Please include relevant statistical difference between the groups in these parameters, that may influence the outcomes and conclusions. Was the receptor expression altered between the groups (so not per subset, but the receptor expression itself between the groups?).

Further, it is not clear whether there was a difference in steroid use between the control and COPD group shown in Figure 2 and SFigure 2 (the validation cohort).

Also, please provide information regarding glucocorticoid receptor expression and responses in myeloid subsets (Fig 5).

In fig 2B,C, the authors state that the upregulation of IL6-STAT3 pathway supports a potential Th17 phenotype in E-COPD. If possible using the current dataset, please provide data on the Th17 cells between the groups.

The authors use Palantir modeling for pseudotemporal analysis of the level of differentiation of the measured CD4 T cell subsets. Please discuss the origin of these subsets: can the authors distinguish resident and recruited subsets, and are these different between the groups (also based on the findings presented in figure 6, related to CXCR6 expression)?

The included subjects in this study are assumedly clinically stable to under go the procedure before tissue collection. As steroid use seems different between groups, could underlying (opportunistic) infections, interfere with the presented data, as IL6-STAT signaling, but also other pathways related to inflammation are increased in E-COPD compared to the other groups (Fig 2B, C).

The authors show the percentage of NR3C1+ subsets of total CD4 in figure 4A, however, it is unclear whether there is a difference in other subsets of CD4, or of CD4 as part of total lymphocytes (although in figure 1 the CD4 population is shown to be increased). Please add these data for completeness.

Furthermore, from Figure 4C, the authors conclude that the NR3C1+ CD4 subset is involved in reducing inflammation. It would be helpful to support this conclusion by measurement of inflammatory parameters in the patients.

Version 1:

Reviewer comments:

Reviewer #1

(Remarks to the Author)

I thank the authors for the work they have performed, which has significantly improved the quality of the manuscript. The responses to my previous comments regarding the scRNA-seq analysis are satisfactory. However, there are still no functional experiments to validate even one of their findings. In their absence, this study remains very descriptive.

Reviewer #2

(Remarks to the Author)

All raised issues were addressed.

Thank you for addressing the comments.

Below, please find our responses to the outlined *comments (i-iv)*, and a point-by-point rebuttal addressing our reviewer's comments:

(i) Exclude the potential influence of significant confounding factors across patient groups, particularly the impact of past and current smoking on immune responses.

Response i: We appreciate the concern regarding smoking status. Former smokers are present across all three groups in our dataset, and the proportions of former smokers are as follows: 9/25 (36%) in controls, 12/21 (57%) in nonemphysematous COPD (NE-COPD), and 16/16 (100%) in emphysematous COPD (E-COPD) group. Because all E-COPD patients are former smokers, we are not able to isolate the effects of active smoking in the E-COPD group. To address the effect of active smoking in the NE-COPD and the Control groups, we further stratified smoking status in both disease groups and assessed cellular enrichments. This analysis is included in the new **Supplemental Figure 2** and shown here as **Rebuttal Figure 1**. We found that in NE-COPD, former smokers exhibit negative enrichment of CD8 T cells but positive enrichment of macrophages (**Rebuttal Figure 1c**). CD4 and CD8 T cells were negatively enriched, and macrophages were positively enriched when comparing current smokers to never-smokers (**Rebuttal Figure 1d**).

Supplemental Figure 2

smokers exhibit negative enrichment of CD8 T cells but positive enrichment of macrophages (**Rebuttal Figure 1c**). CD4 and CD8 T cells were negatively enriched, and macrophages were positively enriched when comparing current smokers to never-smokers (**Rebuttal Figure 1d**).

Rebuttal Figure 1: Positive Enrichment of Macrophages in active smokers in NE-COPD and Control. (a) Individual and (b) collective cellular proportions across never (N=1), former (N=12), and current (N=8) smokers in the NE-COPD group. Pairwise comparison of cell type enrichment using scCODA between (c) former & never-smokers, (d) current & never-smokers, (e) current & former smokers. (f) Individual and (g) collective cellular proportions across never (N=11), former (N=9), and current (N=5) smokers in the control group. Pairwise comparison of cell type enrichment using scCODA between (h) former & never-smokers, (i) current & never-smokers. Positive values of enrichment level suggest positive enrichments and negative values suggest negative enrichments. The significance threshold for false discovery rate (FDR) in scCODA differential abundance analyses was set at 0.25.

Consistent with prior studies,¹ current smokers show macrophage enrichment in the lungs (**Rebuttal Figure 1d**). When comparing current smokers to former smokers, we observed negative enrichment of CD4 T cells (**Rebuttal Figure 1e**). In the control group, comparisons between current and never smokers did not yield any statistically significant enrichment. However, CD4 T cells were positively enriched, while macrophages were negatively enriched in former smokers compared to never smokers (**Rebuttal Figure 1h**). Macrophages and CD8 T cells were found to be positively enriched (**Rebuttal Figure 1i**) in current smokers when compared to former smokers. These results suggest that the effect of active smoking is primarily on pulmonary macrophages.

(ii) Conduct functional experiments to validate the predicted cell differentiation trajectories and cell-cell interactions.

Response ii: We appreciate the comment and agree that future in vivo work could confirm our in-silico analysis. To test the validity of our observation, we used a second dataset (GEO accession: GSE136831) and found that CXCR6 expression classifies effector memory CD4 T cells in both datasets and in trajectory estimation using the unsupervised CellRank² CytoTrace module.³ However, validating the predicted cell differentiation trajectories using lineage tracing transgenic animal models is beyond the scope of the current study. We have revised the discussion point to point out this unmet need for future research in animal models of cigarette-smoke-induced emphysema. Please see lines 373-376.

(iii) Provide detailed information about the types of steroids used and perform statistical analyses to account for potential confounding factors influencing the differential enrichment of NR3C1+ CD4 T cells in NE-COPD and E-COPD. Additionally, compare the overall expression levels of the receptor across groups.

Response iii: We examined the relative abundance of NR3C1+ CD4 subset in subjects that 1) did not use any steroid (None), 2) used inhaled corticosteroids (ICS), 3) used oral corticosteroids (OCS), and 4) used combined inhaled corticosteroids and oral corticosteroids (ICS+OCS) in each cohort. The E-COPD group had the highest rate of corticosteroid usage, and inhaled corticosteroids represent the most prevalent corticosteroid types.

Consistent with our original data, we did not find any significant effects of the types of corticosteroid usage on the relative abundances of NR3C1+ CD4 T cells. This new information is provided in the new **Figure 4c and Rebuttal Figure 2**.

Rebuttal Figure 2. Effects of types of Corticosteroid usage on the relative abundance of NR3C1 CD4 subsets. Percentages of relative abundance of NR3C1 subsets in CD4 are plotted regarding the corticosteroid types used. OCS: Oral corticosteroids; ICS: inhaled corticosteroids; ICS+OCS: combined inhaled and oral corticosteroids.

In response to the second comment, we examined the expression levels of NR3C1 in CD4 T cells in our three cohorts. As expected, we did not find a significant difference across the cohorts (**Rebuttal Figure 3**).

Rebuttal Figure 3. Expression of NR3C1 in CD4 NR3C1 subsets across 3 groups. The y-axis represents logarithmically transformed count data of the NR3C1 gene

(iv) Present data showing the correlation between the NR3C1+ CD4 subset and inflammation.

Response iv: We used different approaches to assess the correlations between the NR3C1+ CD4 subset and inflammation (**Rebuttal Figure 4 & new Figure 6**).

While scRNA-seq is a powerful tool to examine the composition of immune cells in the lungs, this technology is not appropriate to find cellular correlation within the tissue. Therefore, we performed Nanostring GeoMX to assess RNA transcript levels at specific microscopic tissue regions to establish the association of different innate immune cells as representative of the inflammatory cell populations within the lung tissue. We then deconvoluted the bulk RNA transcript levels of the parenchyma regions into specific cell types identified using the in-house single-cell RNA-sequencing of defined cell types with CIBERTSORTx.⁴ Our spatial transcriptomic data showed that in NE-COPD, the relative abundance of NR3C1+ CD4 positively correlates with neutrophils and migratory dendritic cells, but again, such a correlation was not observed in E-COPD (**Rebuttal Figure 4a**).

Next, we assessed the plasma chemokine/cytokine levels for patients in the in-house single-cell RNA-sequencing dataset using multiplexed cytokine/chemokine beads-based array and correlated the relative abundance of CD4 subsets. We found no significant correlation in NE-COPD, but in E-COPD, CXCL13, CXCL5,

and CXCL1 (highlighted in red) positively correlated with the relative abundance of NR3C1⁺ CD4 T. CXCL13 is a major chemokine for germinal center formation ⁵. Both CXCL5 ⁶ and CXCL1 ⁷ are involved in neutrophil chemotaxis. (**Rebuttal Figure 4b**). These results suggest that NR3C1⁺ CD4 T might be engaged in the lymphoid follicle development and maintenance, as well as neutrophil recruitment in E-COPD.

Rebuttal Figure 4. Correlation of NR3C1 CD4 subset and inflammation. (a) Relative cellular abundance of Nanostring GeoMX data in the lung parenchyma was deconvoluted using in-house single-cell RNA sequencing cell type signatures as references. Spearman correlations of deconvoluted relative abundances were performed for NE-COPD and E-COPD groups. **(b)** Spearman correlation of the relative abundance for CD4 subsets and plasma cytokine/chemokine levels in NE-COPD and E-COPD. Significance code: *p<0.05; **p<0.01; ***p<0.001

Reviewer #1 (Remarks to the Author):

General Comments:

In the manuscript entitled « Lung NR3C1+ and CXCR6high T cells distinguish immunopathogenesis of human emphysema », the authors propose to explore the specific alteration of the immune system in human emphysema, using single cell-RNA analysis performed on lung samples. The strengths of this study are the number of clinical samples (in total 62 tissue samples), their exhaustive clinical characterization (in particular radiographic quantification emphysema) as well as the careful and precise scRNA seq analysis. Notably, the authors also use a validation cohort (64 samples), from a publicly available scRNA seq dataset) to validate their findings. The main results include the identification of two subsets of immune cells associated with emphysema: NR3C1+CD4 T cells, reduced in emphysematous patients, and pro-inflammatory CD4 effector memory cells, increased in emphysematous patients. Main weaknesses are the following:

In addition of the presence of emphysema and airflow obstruction, there are (too) many confounding factors in the three different groups (age, current/former smoker and treatment) that could strongly affect the results. Past and current smoking is especially problematic, as it is well known to affect immune responses.

Response: We thank the reviewer for this comment. We acknowledge that smoking status can be a major confounding factor. Because all patients in E-COPD are former smokers, we have stratified the NE-COPD and the Control groups by smoking status and assessed the effects of smoking on pulmonary cellular composition. These results are shown in **Rebuttal Figure 1** and the updated **Supplemental Figure 2**. We have found that the primary effect of active smoking is on pulmonary macrophages in both the Control and the NE-COPD groups (**Rebuttal Figure 1d and 1i**).

Moreover, this study is very descriptive. Most of the « mechanistic » conclusions were drawn on pseudotemporal analysis and ligand-receptor analysis to predict cell differentiation trajectories and cell-cell interactions, respectively. Both types of analyses are based on tools that have serious limitations and bias. In the absence of any functional experiments, or at least spatial cellular analysis, the conclusions remain very speculative and do not bring much on the understanding of emphysema.

Response: We appreciate the comments, and while human translational studies are primarily hypothesis-generating, they are critical in advancing our current understanding of the pathophysiology of human diseases. As such, we conducted our studies using a large cohort of well-characterized normal controls, smokers with and without emphysema, to identify T cell signatures that could inform on how adaptive immune cells can persist in the lungs of smokers. In response to this concern, we have expanded on the limitations of our studies, which are outlined in the discussion (please see lines 380-384).

The authors focused on late stages of emphysema, which is another limitation, as it only shows the “final picture”. The inclusion of another group of patients, with emphysematous tissue destruction without major lung function alteration would have represented a significant advance to identify cellular alterations at early stages, to propose putative cellular culprits to initiate and propagate emphysematous lesions.

Response: We agree with the reviewer that a large cohort of emphysema patients with different stages of disease would be ideal to ask new questions. Nonetheless, published literature investigating COPD in humans often profiles the immune status of the peripheral blood, and human COPD studies rarely isolate emphysema but instead only focus on pulmonary function-based classification of COPD. In the present study, we compared pulmonary tissue immunity between nonemphysematous and emphysematous COPD and found that emphysematous COPD exhibits a distinct pulmonary immune profile compared to nonemphysematous COPD. We further found that CD4 effector memory correlates with lung function in only emphysematous COPD but not nonemphysematous COPD and control groups. Our results emphasize the need for stratification of chest computed tomography (CT)-defined emphysema in the investigation of COPD, in addition to pulmonary function tests. Pulmonary function tests remain the mainstream diagnostic tool for COPD, and chest CT scans are often reserved for those with advanced diseases or patients in need of surgery. Future studies may focus on collecting patients with emphysematous pulmonary pathology with or without airflow obstruction to further isolate the immune contribution in the destruction of lung tissue in emphysema.

Major points:

#1. The control and COPD tissue samples did not come from the same type of sampling (medically necessary surgical explants for control tissues, lung volume reduction or transplant surgeries for COPD tissues). How did the authors control this potential major bias?

Response 1: Human lung tissues are required for the profiling of pulmonary tissue immunity, but it is noteworthy that the procurement of human tissues is logistically challenging. For controls, we used samples from human donor lungs that were not used in a transplant. We only use distal sections of the excised lung tissue and collected tissues that were at least 10 cm away from the tumor or non-involved lobes, as described in the Materials and Methods section “In-house Dataset One”. We have previously addressed concerns regarding immune cells harvested from lung volume reduction surgery compared to medically necessary lung resections. We showed that the lung immune profile in tissues resected 10 cm away from the lung tumor does not differ from samples collected from lung volume reduction surgery cases.⁸

#2. The sc-analysis is performed on “lung tissue”. Given that emphysema is a specific alteration of alveoli, we would have expected a selection of the distal part of the lungs. Could the authors precise if such a selection was performed? If not, this should be included in the discussion as a limitation.

Response 2: We only used distal sections of the excised human lung tissue, as described in **Response 1**.

#3. There are many factors of variations between the three groups of patients: 1) the median age of the 3 different groups of patients is different: control patients are on average younger than the patients with COPD, and NE-COPD patients seem also older than E-COPD patients. 2) the percentage of current vs former smokers is also very different in the 3 groups. 3) the treatment (steroids) also differs between NE and E-COPD patients. How did the authors control these three potential major biases?

Response 3: We appreciate this concern; however, as described in **Response 1**, normal human lung tissue can only be obtained in cases of “donor lung”, as we and others have used.⁸ Therefore, the age of the participant could not be controlled for in human tissue-based studies. In addition, most patients were above 50 years of age, and no statistically significant differences were observed between Control vs E-COPD and NE-COPD vs E-

COPD (**Supplemental Figure 1d**). In response to the second confounder concern regarding smoking status, we now provide new data using scRNA-seq from former, never smokers, and active smokers in our NE-COPD and Control cohort (**Rebuttal Figure 1** and **Supplemental Figure 2**). As all subjects in E-COPD are former smokers, we were not able to isolate the effects of smoking in the E-COPD group. We found that active smoking positively enriches macrophages when compared to never smokers (**Rebuttal Figure 1**). We have also examined the effects of inhaled corticosteroids and oral corticosteroids on NR3C1 CD4 T cells in NE-COPD and E-COPD groups. Please see **Rebuttal Figure 2 and the updated Figure 4c**.

#4. Quality control of single-cell data: The authors provided a total number of cells that passed quality control: “a total of 57,223 cells from the control, 55,632 cells from NE-COPD, and 46,847 cells from the E-COPD group passed quality control.” Could the authors provide the same quantifications per biological sample, to test for interindividual variations?

Response 4: In response to this request, we now provide individual cell recovery for subjects across three groups. To address whether the differences we observed might be due to the differences in cellular recovery efficiency between different groups, we compared the cell number recovered in each patient and compared recovery efficiency between the three groups which did not show any statistically significant differences (**Rebuttal Figure 5**).

Rebuttal Figure 5. Cell recovery efficiency across 3 disease groups. Total cell number recovered in the individual patient was plotted for all three disease groups. The Kruskal-Wallis test was used to determine whether the three disease groups derive from the same distribution for cellular recovery. A pairwise comparison was performed using the Wilcoxon rank-sum test with Holm’s correction. ns: non-significant.

#5. The validation cohort is problematic: as the information regarding %LAA data was not available, there is only distinction between COPD and control patients. Then the validation of specific alterations in emphysematous patients identified with the original cohort is not possible.

Response 5: We appreciate the concern; however, datasets that are publicly accessible do not have metadata including disease severity, %LAA, steroid treatment status, etc. Despite these challenges, we were able to identify the same distinct subset of CD4 T cells in the lungs, validating a major aspect of our findings. Indeed, most human COPD-based studies group patients into different strata using GOLD staging, which is based on PFTs, without the use of %LAA. Therefore, while we could not find access to the validation dataset with detailed metadata, our current work provides this important unmet need.

#6. Figure 1e “epithelial cells showed negative enrichments in E-COPD compared to controls”. This seems to be due to a loss of airway epithelial cells, but not of alveolar epithelial cells. Can the authors comment on the fact that no alveolar epithelial loss was detected in the emphysematous condition in comparison to the other ones?

Response 6: We used single-cell RNA sequencing, which was optimized for immune cell profiling. Further, the tissue dissociation required to prepare a single-cell suspension often enriches immune cells compared to a single-nucleus RNA sequencing-based protocol^{9,10}. Therefore, in this report, we did not focus on potential changes in alveolar epithelial cells, but given the importance of cell-cell interaction, it will be a subject of our future studies.

#7. Figure 2: “IL6 induces expression of IL21 and IL23R in CD4 T cells, which are upstream of RORC and IL17 expression, and STAT3 is also indispensable in the development of Th17 cells”: are Th17 cells identified in scRNA data, and how are they modified in the different groups of patients?

Response 7: We examined the expression of IL-17A and RORC, however, in the CD4 T cell population, sequencing depth in this population was not sufficient for their direct identification. To circumvent this technical issue, we examined the upstream pathways (e.g., IL-6, IL-21, and IL-23R) in CD4 T cells that indicate activation of this subset as described previously¹¹⁻¹³. Furthermore, in response to this concern, we also

scored for the IL-17A production pathway in the gene ontology biological process (GOBP) using Gene Set Variation Analysis (GSVA). Consistent with the upstream IL-17A pathways, E-COPD showed increased IL-17A production signaling compared to Controls (**Rebuttal Figure 6**). This information is now added to the updated **Figure 2**.

Rebuttal Figure 6. GSVA score of IL17 production pathways (Gene Ontology, Biological Process, Accession GO: 0032620) for each patient across three disease cohorts. Heterogeneity between the three groups was first determined using the Kruskal-Wallis test and pairwise comparisons were performed using the Wilcoxon rank-sum test. ns: non-significant; **: p<0.01.

#8. The identification of the specific subset of CD4 T cells expressing glucocorticoid receptor NR3C1 and its reduction in E-COPD patients is interesting. The authors claimed that the “alterations in the relative abundance of the NR3C1 subset were independent of individual patient’s glucocorticoid therapy (Figure 4b)”. However, although it is not statistically significant, it seems that there is a higher level of NR3C1 positive cells in CD4+ cells from patients with steroids treatment. Could the authors comment on this point, and also provide a similar analysis than the one provided in Figure 4b, but with NR3C1 level of expression?

Response 8: We thank this reviewer for their careful observation. In response to this concern, we examined oral and inhaled glucocorticoid usage in our E-COPD and NE-COPD cohorts and found no significant effects of the types of steroids used and the relative abundance of NR3C1 CD4 T cells (**Rebuttal Figure 2** and updated **Figure 4c**). In response to the second part of the concern, we examined the expression of NR3C gene levels across three cohorts and did not find any significant differences (**Rebuttal Figure 3**).

#9. Interactome modeling: based on transcriptional levels of ligand-receptor pairs, intercellular interactions are predicted, such as the interplay between pulmonary macrophages and NR3C1+ CD4 T cell subset in E-COPD patients. While this could be a good starting point, this is clearly not sufficient, in particular because transcriptional levels do not always reflect protein levels nor the proximity of the cells. The relevance of the interactions identified should be validated by quantifying protein levels and by spatially studying the relative distribution of implicated cells.

Response 9: In response to this concern, we analyzed Nanostring GeoMX spatial transcriptomics data using lung tissue from the same cohort to assess cellular colocalization. We deconvoluted the spatial data with CiberSORTx using the cell type signature from the in-house single-cell RNA sequencing dataset as reference. We examined the cellular proportions in lymphoid follicles and parenchyma in the Nanostring GeoMX dataset and found that Migratory DC and CXCR6^{High} CD4 effector memory, CD4 NR3C1 and PPARG macrophages

exhibit similar enrichment patterns in different tissue compartments. Interestingly, we also found that while in NE-COPD, PPARG macrophages and NR3C1⁺ CD4 T cells are enriched in parenchyma, in E-COPD, PPARG macrophages and NR3C1⁺ CD4 T cells are enriched in the lymphoid follicles. CD4 CXCR6^{High} effector memory and Migratory DCs were enriched in lymphoid follicles in both NE-COPD and E-COPD groups. Furthermore, published animal studies have demonstrated that tissue macrophages interact with T cells by secreting endogenous glucocorticoids¹⁴. (**Rebuttal Figure 7**)

Rebuttal Figure 7. Relative distribution of immune cells in lymphoid follicles and parenchyma in the human lungs. Nanostring

GeoMX dataset was deconvoluted with CiberSORTx using in-house single cell RNA sequencing cell type signatures as reference. Relative distribution of identified immune cells in lung lymphoid follicles and lung parenchyma were assessed in NE-COPD (a) and E-COPD(b).

#10. P9 *“Together, these results suggest that higher expressions of immune checkpoints in CXCR6^{High} CD4 Effector memory might contribute to the preserved lung function in E-COPD.” This conclusion is very speculative, as it is based on correlations. The authors propose that the underlying mechanism is the interaction between CXCR6 high CD4 effector memory and migratory DC, based on interactome analysis. Can the authors provide data to support this hypothesis? Functional experiments are here required.*

Response 10: To examine the relative distribution of migratory dendritic cells and CXCR6^{High} CD4 effector memory T cells, we analyzed our unpublished Nanostring GeoMX dataset by deconvoluting cell types with CiberSORTx using our single-cell RNA sequencing dataset cell type signatures as reference. These results are shown in **rebuttal Figure 7**.

#11. Discussion *“emphysematous tissue destruction can be found in some smokers with chest computed tomography (CT) in the absence of any detectable PFT abnormalities”:* in fact, including these samples would be really interesting to look at early stages of emphysema development.

Response 11: We agree with this reviewer and have expanded this point in the discussion because we agree that early-stage emphysema should be studied in similar depth as we have done here. Please see lines 381-385 in the discussion.

Reviewer #2 (Remarks to the Author):

In this paper, the authors applied single cell RNA sequencing on a large number of clinical samples, with detailed clinical information, and validated the findings in an online available database. The authors show that subsets of CD4 lymphocytes, expressing NR3C1 and CXCR, distinguish emphysema-COPD from COPD without emphysema. The validation cohort strengthen the findings of the authors. Furthermore, the authors provide detailed clinical data and link the single-cell RNA seq and interactome data to the clinical findings. Combined, the results provide detailed information and pinpoint potential targets for treatment of COPD-emphysema using elegant methods. There are some minor issues that can be addressed to improve the paper and strengthen the findings further.

1) In figure 1D, E there not seem to be a difference in the AT1/AT2 cells between the groups, and in Figure 1D, there is also no difference between COPD and control in airway epithelia. The authors do mentions that the number of cells derived from the individual groups was different. Was there a difference in the number of immune versus structural cells between the groups? In other words, was the emphysema-phenotype reflected on the number of immune/ structural cells? It may be informative to determine whether the LAA% correlates to any of the measured parameters, to determine whether emphysema severity is related to the subsets found.

Response 1: We appreciate this comment and would like to clarify that live-cell single-cell RNA sequencing technology in our in-house cohort was optimized for CD45+ immune cells, which is the focus of the current study. Non-immune cells represent less than 10% of the recovered cells in our in-house dataset, consistent with published empirical observations.^{9,10} In future studies, we plan to use a different approach that would allow for such direct comparisons.

2) On page 5, lines 2-4, the authors state that the CD8 T cell populations were enriched in E-COPD compared to NE-COPD, but not compared to controls. This seems contradictory, especially since several studies showed that CD8 T cells subsets are increased compared to controls. Please discuss these findings in relation to previously published findings showing that CD8 are increased in emphysema compared to control.

Response 2: We appreciate this comment and, in response, have revised the discussion to include a recent publication showing that CD8 T cells were increased in the early stages of COPD ¹⁵. This information is now included in the discussion (please see lines 382-383).

3) *The expression of NR3C1 in the 2 cohorts seems different (Fig 3A, B versus Supplemental figure 3A, B). Whereas the CD4 subset in Fig 3A is very clearly separated from the other subsets (no/low expression of S110A4 and IL32), the cluster is not very distinguishable in the validation cohort (SFig 3A,B). Please explain this difference. Furthermore, it would be helpful to depict also the source of cells (sample code) between the 3 (for the study cohort, control NE-COPD or E-COPD) or 2 (for the validation cohort) in figure S3A, Fig 3A, but also in Fig 5A, 6A, and S8A.*

Response 3: We appreciate this comment and agree that although we found the same five distinct subsets of T cells in the validation cohort, they are less distinct than the in-house dataset T cell subsets. This could be due to the differences in the enzymatic dissociation for the preparation of single-cell suspension, the chemistry used for scRNA-sequencing library preparation, and the sequencing depth in the two datasets. Our in-house dataset was generated using 5' chemistry, whereas the validation dataset was generated using 3' chemistry from 10XChromium. The validation dataset used elastase and Liberase for tissue dissociation, and for the in-house dataset, collagenase was used for tissue dissociation and preparation of lung single cell suspension. Despite the differences presented in the UMAP embeddings, it is worth noting that both classifications followed similar signature genes for each subset, as shown in **Figure 3c** and **Supplemental Figure 3c**.

4) *The authors state that CD4 T cells expressing the glucocorticoid receptor NR3C1 are enriched in NE-COPD, but reduced in E-COPD, which is one of the main messages of this paper. However, as shown in SFigure 1G, there was a significant difference in the steroid use between these groups, which may modulate expression of the receptor of interest (or localization of the receptor). Also gender and smoking history seems different between the groups. Although authors state that glucocorticoid use did not affect the outcomes, please provide some more information on the types of steroids used. Please include relevant statistical difference between the groups in these parameters, that may influence the outcomes and conclusions. Was the receptor expression altered between the groups (so not per subset, but the receptor expression itself between the groups?).*

Response 4: We thank the reviewer for this comment. We have now provided additional information on inhaled vs oral steroid use in each of our cohorts (please see our **Response iii** to editors and **Response 8** to **Reviewer 1, Rebuttal Figure 4**). We further examined the expression of NR3C1 in CD4 T cells across three disease groups and did not observe any statistical differences across the three disease cohorts (**Rebuttal Figure 3**). We have also examined the effect of smoking, and the results are shown in **rebuttal Figure 1**.

5) *Further, it is not clear whether there was a difference in steroid use between the control and COPD group shown in Figure 2 and SFigure 2 (the validation cohort).*

Response 5: We appreciate this question; however, in the validation cohort from publicly available data, we did not have access to many important metadata, including treatment history, and the history of steroid usage in the validation data is not available (GEO accession: GSE136831).

6) *Also, please provide information regarding glucocorticoid receptor expression and responses in myeloid subsets (Fig 5).*

Response 6: We appreciate this question. In our in-house dataset, we detected robust glucocorticoid receptor NR3C1 expression in pulmonary macrophages. Among all the macrophage subsets, PPARG exhibits the highest levels of expression of NR3C1 and the glucocorticoid receptor co-receptor NCOA1 (**Rebuttal Figure 8**).

Rebuttal Figure 8. Expression of NR3C1 in macrophage subsets.

(a) Dotplot of NR3C1 and NCOA1 expression across macrophage subsets. Violin plots of (b) NR3C1 and (c) NCOA1 expression across macrophage subsets. The y-axis represents logarithmically transformed count data of the NR3C1 gene for each cell.

7) In fig 2B,C, the authors state that the upregulation of IL6-STAT3 pathway supports a potential Th17 phenotype in E-COPD. If possible, using the current dataset, please provide data on the Th17 cells between the groups.

Response 7: In both the in-house dataset and validation datasets, lymphocytes sequencing depth did not allow for identification of key Th17-associated genes such as IL17A and RORC. However, we scored the Gene Ontology pathway “IL17 production” in CD4 T cells across three disease groups and found augmented IL17 production in CD4 T cells in E-COPD compared to controls (**Rebuttal Figure 6**).

8) The authors use Palantir modeling for pseudotemporal analysis of the level of differentiation of the measured CD4 T cell subsets. Please discuss the origin of these subsets: can the authors distinguish resident and recruited subsets, and are these different between the groups (also based on the findings presented in figure 6, related to CXCR6 expression)?

Response 8: We thank our reviewer for this insightful comment. In response to this question, we examined the expression of CD103 (ITGAE) a distinct marker associated with resident effector memory T cells¹⁶, in the CD4 effector memory T cells (CCR7⁻ CXCR6^{high}) subset. We found that the majority of CD4 T cells in the CCR7⁻ CXCR6^{high} subset express the highest level CD103, indicating that they are likely to be tissue-resident (**Rebuttal Figure 9**).

Rebuttal Figure 9. Residency marker expression across all identified CD4 subsets.

9) The included subjects in this study are assumedly clinically stable to under go the procedure before tissue collection. As steroid use seems different between groups, could underlying (opportunistic) infections, interfere with the presented data, as IL6-STAT signaling, but also other pathways related to inflammation are increased in E-COPD compared to the other groups (Fig 2B, C).

Response 9: We are unable to directly respond to this interesting question as we did not assess tissue microbiome, and no infectious history has been included in the metadata at this time. In future studies, a correlational analysis of lung microbiota could address this question.

10) The authors show the percentage of NR3C1+ subsets of total CD4 in figure 4A, however, it is unclear whether there is a difference in other subsets of CD4, or of CD4 as part of total lymphocytes (although in figure 1 the CD4 population is shown to be increased). Please add these data for completeness.

Response 10: We thank the reviewer for the constructive criticism. We have shown the relative abundance of other CD4 subsets in **Rebuttal Figure 7**. Among all the subsets identified, only the NR3C1⁺ CD4 subset showed a statistically significant reduction in proportions in E-COPD when compared to NE-COPD (**Rebuttal**

Figure 10e).

Rebuttal Figure 10. Relative abundance of CD4 subsets across three disease groups. Relative abundances of CD4 subsets for individual patients are plotted against disease status. Statistical significance was first determined using Kruskal-Wallis test and pairwise comparisons were performed using Wilcoxon rank sum tests with Holm's correction for multiple comparisons.

11) Furthermore, from Figure 4C, the authors conclude that the NR3C1+ CD4 subset is involved in reducing inflammation. It would be helpful to support this conclusion by measurement of inflammatory parameters in the patients.

Response 11: We thank our reviewer for their careful review of our data. In Figure 4, we show that NE-COPD shows an elevated relative abundance of NR3C1+ CD4 T cells. Our subsequent analysis mapping of NR3C1 targets showed that pro-inflammatory IL6, RELA, and JUN are among the inhibited targets, while the signature genes of the NR3C1 CD4 subset to pathway gene sets showed that TGFβ signaling is among the top upregulated pathways, suggesting an anti-inflammatory function of this subset (**Figure 4d**). Together, these findings suggest a potential role of this subset of T cells in reducing inflammation. In addition, we correlated the relative abundance of NR3C1 CD4 T cells with plasma cytokine/chemokine levels, shown in **rebuttal Figure 4 and new Figure 6**. We have found that NR3C1 CD4 T cells positively correlate with CXCL13, CXCL5, and CXCL1. CXCL13 is a major chemokine for germinal center formation ⁵. Both CXCL5 ⁶ and CXCL1 ⁷ are involved in neutrophil chemotaxis. These results suggest a more complicated role of NR3C1 CD4 T cells in emphysema development and progression. We have revised the text in the manuscript to clarify this point (please see lines 216-233).

References:

- Hodge, S., et al. Smoking alters alveolar macrophage recognition and phagocytic ability: implications in chronic obstructive pulmonary disease. *Am J Respir Cell Mol Biol* **37**, 748-755 (2007).
- Lange, M., et al. CellRank for directed single-cell fate mapping. *Nat Methods* **19**, 159-170 (2022).
- Gulati, G.S., et al. Single-cell transcriptional diversity is a hallmark of developmental potential. *Science* **367**, 405-411 (2020).
- Newman, A.M., et al. Robust enumeration of cell subsets from tissue expression profiles. *Nat Methods* **12**, 453-457 (2015).
- Havenar-Daughton, C., et al. CXCL13 is a plasma biomarker of germinal center activity. *Proc Natl Acad Sci U S A* **113**, 2702-2707 (2016).
- Mei, J., et al. Cxcr2 and Cxcl5 regulate the IL-17/G-CSF axis and neutrophil homeostasis in mice. *J Clin Invest* **122**, 974-986 (2012).

7. De Filippo, K., *et al.* Mast cell and macrophage chemokines CXCL1/CXCL2 control the early stage of neutrophil recruitment during tissue inflammation. *Blood* **121**, 4930-4937 (2013).
8. Grumelli, S., *et al.* An immune basis for lung parenchymal destruction in chronic obstructive pulmonary disease and emphysema. *PLoS Med* **1**, e8 (2004).
9. Slyper, M., *et al.* Author Correction: A single-cell and single-nucleus RNA-Seq toolbox for fresh and frozen human tumors. *Nat Med* **26**, 1307 (2020).
10. Wen, F., Tang, X., Xu, L. & Qu, H. Comparison of single- nucleus and single- cell transcriptomes in hepatocellular carcinoma tissue. *Mol Med Rep* **26**(2022).
11. Pawlak, M., *et al.* Induction of a colitogenic phenotype in Th1-like cells depends on interleukin-23 receptor signaling. *Immunity* **55**, 1663-1679 e1666 (2022).
12. Zhang, C.J., *et al.* Act1 is a negative regulator in T and B cells via direct inhibition of STAT3. *Nat Commun* **9**, 2745 (2018).
13. Lee, Y., *et al.* IL-21R signaling is critical for induction of spontaneous experimental autoimmune encephalomyelitis. *J Clin Invest* **125**, 4011-4020 (2015).
14. Shimba, A., *et al.* Glucocorticoids Drive Diurnal Oscillations in T Cell Distribution and Responses by Inducing Interleukin-7 Receptor and CXCR4. *Immunity* **48**, 286-298 e286 (2018).
15. Villasenor-Altamirano, A.B., *et al.* Activation of CD8(+) T Cells in Chronic Obstructive Pulmonary Disease Lung. *Am J Respir Crit Care Med* **208**, 1177-1195 (2023).
16. Hirahara, K., Kokubo, K., Aoki, A., Kiuchi, M. & Nakayama, T. The Role of CD4(+) Resident Memory T Cells in Local Immunity in the Mucosal Tissue - Protection Versus Pathology. *Front Immunol* **12**, 616309 (2021).